# Accurate Learning of Graph Representations with Graph Multiset Pooling

**Jinheon Baek**[1*], **Minki Kang**[1*], **Sung Ju Hwang**[1,2]
KAIST[1], AITRICS[2], South Korea
{jinheon.baek, zzxc1133, sjhwang82}@kaist.ac.kr

## Abstract

Graph neural networks have been widely used on modeling graph data, achieving impressive results on node classification and link prediction tasks. Yet, obtaining an accurate representation for a graph further requires a pooling function that maps a set of node representations into a compact form. A simple sum or average over all node representations considers all node features equally without consideration of their task relevance, and any structural dependencies among them. Recently proposed hierarchical graph pooling methods, on the other hand, may yield the same representation for two different graphs that are distinguished by the Weisfeiler-Lehman test, as they suboptimally preserve information from the node features. To tackle these limitations of existing graph pooling methods, we first formulate the graph pooling problem as a multiset encoding problem with auxiliary information about the graph structure, and propose a *Graph Multiset Transformer* (GMT) which is a multi-head attention based global pooling layer that captures the interaction between nodes according to their structural dependencies. We show that GMT satisfies both injectiveness and permutation invariance, such that it is at most as powerful as the Weisfeiler-Lehman graph isomorphism test. Moreover, our methods can be easily extended to the previous node clustering approaches for hierarchical graph pooling. Our experimental results show that GMT significantly outperforms state-of-the-art graph pooling methods on graph classification benchmarks with high memory and time efficiency, and obtains even larger performance gain on graph reconstruction and generation tasks.[1]

## 1 Introduction

Graph neural networks (GNNs) (Zhou et al., 2018; Wu et al., 2019), which work with graph structured data, have recently attracted considerable attention, as they can learn expressive representations for various graph-related tasks such as node classification, link prediction, and graph classification. While the majority of the existing works on GNNs focus on the message passing strategies for neighborhood aggregation (Kipf & Welling, 2017; Hamilton et al., 2017), which aims to encode the nodes in a graph accurately, graph pooling (Zhang et al., 2018; Ying et al., 2018) that maps the set of nodes into a compact representation is crucial in capturing a meaningful structure of an entire graph.

As a simplest approach for graph pooling, we can average or sum all node features in the given graph (Atwood & Towsley, 2016; Xu et al., 2019) (Figure 1 (B)). However, since such simple aggregation schemes treat all nodes equally without considering their relative importance on the given tasks, they can not generate a meaningful graph representation in a task-specific manner. Their flat architecture designs also restrict their capability toward the hierarchical pooling or graph compression into few nodes. To tackle these limitations, several differentiable pooling operations have been proposed to condense the given graph. There are two dominant approaches to pooling a graph. Node drop methods (Zhang et al., 2018; Lee et al., 2019b) (Figure 1 (C)) obtain a score of each node using information from graph convolutional layers, and then drop unnecessary nodes with lower scores at each pooling step. Node clustering methods (Ying et al., 2018; Bianchi et al., 2019) (Figure 1 (D)), on the other hand, cluster similar nodes into a single node by exploiting their hierarchical structure.

---

[*]Equal contribution
[1]Code is available at https://github.com/JinheonBaek/GMT

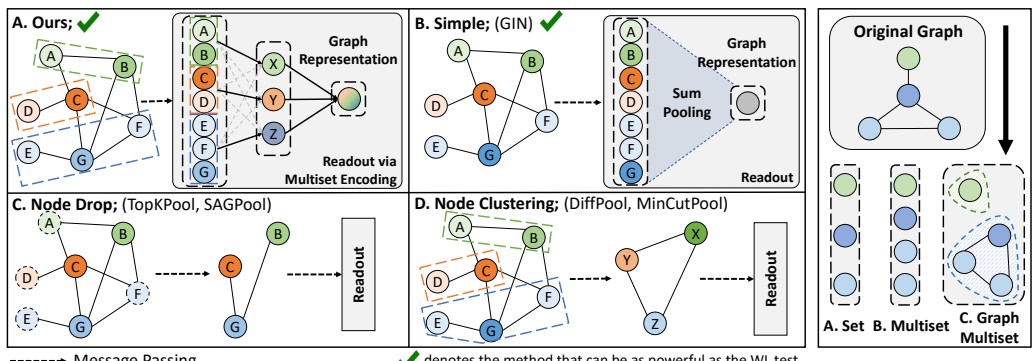

Figure 1: **Concepts (Left):** Conceptual comparison of graph pooling methods. Grey box indicates the readout layer, which is compatible with our method. Also, green check icon indicates the model that can be as powerful as the WL test. **(Right):** An illustration of set, multiset, and graph multiset encoding for graph representation.

Both graph pooling approaches have obvious drawbacks. First, node drop methods unnecessarily drop some nodes at every pooling step, leading to information loss on those discarded nodes. On the other hand, node clustering methods compute the *dense* cluster assignment matrix with an adjacency matrix. This prevents them from exploiting sparsity in the graph topology, leading to excessively high computational complexity (Lee et al., 2019b). Furthermore, to accurately represent the graph, the GNNs should obtain a representation that is as powerful as the Weisfeiler-Lehman (WL) graph isomorphism test (Weisfeiler & Leman, 1968), such that it can map two different graphs onto two distinct embeddings. While recent message-passing operations satisfy this constraint (Morris et al., 2019; Xu et al., 2019), most deep graph pooling works (Ying et al., 2018; Lee et al., 2019b; Gao & Ji, 2019; Bianchi et al., 2019) overlook graph isomorphism except for a few (Zhang et al., 2018).

To obtain accurate representations of graphs, we need a graph pooling function that is as powerful as the WL test in distinguishing two different graphs. To this end, we first focus on that the graph representation learning can be regarded as a *multiset* encoding problem, which allows for possibly repeating elements, since a graph may have redundant node representations (See Figure 1, right). However, since a graph is more than a multiset due to its structural constraint, we further define the problem as a *graph multiset* encoding, whose goal is to encode two different graphs, given as multisets of node features with auxiliary structural dependencies among them (See Figure 1, right), into two unique embeddings. We tackle this problem by utilizing a graph-structured attention unit. By leveraging this unit as a fundamental building block, we propose the *Graph Multiset Transformer* (GMT), a pooling mechanism that condenses the given graph into the set of representative nodes, and then further encodes relationships between them to enhance the representation power of a graph. We theoretically analyze the connection between our pooling operations and WL test, and further show that our graph multiset pooling function can be easily extended to node clustering methods.

We then experimentally validate the **graph classification** performance of GMT on 10 benchmark datasets from biochemical and social domains, on which it significantly outperforms existing methods on most of them. However, since graph classification tasks only require discriminative information, to better quantify the amount of information about the graph in condensed nodes after pooling, we further validate it on **graph reconstruction** of synthetic and molecule graphs, and also on two **graph generation** tasks, namely molecule generation and retrosynthesis. Notably, GMT outperforms baselines with even larger performance gap on graph reconstruction, which demonstrates that it learns meaningful information without forgetting original graph structure. Finally, it improves the graph generation performance on two tasks, which shows that GMT can be well coupled with other GNNs for graph representation learning. In sum, our main contributions are summarized as follows:

- We treat a graph pooling problem as a multiset encoding problem, under which we consider relationships among nodes in a set with several attention units, to make a compact representation of an entire graph only with one global function, without additional message-passing operations.

- We show that existing GNN with our parametric pooling operation can be as powerful as the WL test, and also be easily extended to the node clustering approaches with learnable clusters.

- We extensively validate GMT for graph classification, reconstruction, and generation tasks on synthetic and real-world graphs, on which it largely outperforms most graph pooling baselines.

## 2 RELATED WORK

**Graph Neural Network** Existing graph neural network (GNN) models generally encode the nodes by aggregating the features from the neighbors (Kipf & Welling, 2017; Hamilton et al., 2017; Velickovic et al., 2018; You et al., 2019), and have achieved a large success on node classification and link prediction tasks. Recently, there also exist transformer-based GNNs (Nguyen et al., 2019; Rong et al., 2020) that further consider the relatedness between nodes in learning the node embeddings. However, accurately representing the given graph as a whole remains challenging. While using mean or max over the node embeddings allow to represent the entire graph for graph classification (Duvenaud et al., 2015; Dai et al., 2016), they are mostly suboptimal, and may output the same representation for two different graphs. To resolve this problem, recent GNN models (Xu et al., 2019; Morris et al., 2019) aim to make the GNNs to be as powerful as the Weisfeiler-Lehman test (Weisfeiler & Leman, 1968) in distinguishing graph structures. Yet, they also rely on simple operations, and we need a more sophisticated method to represent the entire graph.

**Graph Pooling** Graph pooling methods play an essential role of representing the entire graph. While averaging all node features is directly used as simplest pooling methods (Atwood & Towsley, 2016; Simonovsky & Komodakis, 2017), they result in a loss of information since they consider all node information equally without considering key features for graphs. To overcome this limitation, there have been recent studies on graph pooling to compress the given graph in a task specific manner. Node drop methods use learnable scoring functions to drop nodes with lower scores (Zhang et al., 2018; Gao & Ji, 2019; Lee et al., 2019b). Moreover, node clustering methods cast the graph pooling problem into the node clustering problem to map the nodes into a set of clusters (Ying et al., 2018; Ma et al., 2019; Wang et al., 2019; Bianchi et al., 2019; Yuan & Ji, 2020). Some methods combine these two approaches by first locally clustering the neighboring nodes, and then dropping unimportant clusters (Ranjan et al., 2020). Meanwhile, edge clustering gradually merges nodes by contracting high-scoring edges between them (Diehl, 2019). In addition, Ahmadi et al. (2020) model the memory layer to aggregate nodes without utilizing message-passing after pooling. Finally, there exists a semi-supervised pooling method (Li et al., 2019) that scores nodes with an attention scheme (Bahdanau et al., 2015), to weight more on the important nodes on pooling.

**(Multi-)Set Representation Learning** Note that a set of nodes in a graph forms a multiset (Xu et al., 2019); a set that allows possibly repeating elements. Therefore, contrary to the previous set-encoding methods, which mainly consider non-graph problems (Qi et al., 2017a; Yi et al., 2019; Snell et al., 2017), we regard the graph representation learning as a multi-set encoding problem. Mathematically, Zaheer et al. (2017); Qi et al. (2017b) provide the theoretical grounds on permutation invariant functions for the set encoding. Further, Lee et al. (2019a) propose Set Transformer, which uses attention mechanism on the set encoding. Building on top of these theoretical grounds on set, we propose the multiset encoding function that explicitly considers the graph structures.

## 3 GRAPH MULTISET POOLING

We posit the graph representation learning problem as a multiset encoding problem, and then utilize the graph-structured attention to consider the global graph structure when encoding the given graph.

### 3.1 PRELIMINARIES

We begin with the general descriptions of graph neural network, and graph pooling.

**Graph Neural Network** A graph $G$ can be represented by its adjacency matrix $\boldsymbol{A} \in \{0, 1\}^{n \times n}$ and the node set $\mathcal{V}$ with $|\mathcal{V}| = n$ nodes, along with the $c$ dimensional node features $\boldsymbol{X} \in \mathbb{R}^{n \times c}$. Graph Neural Networks (GNNs) learn feature representation for different nodes using neighborhood aggregation schemes, which are formalized as the following **Message-Passing** function:

$$\boldsymbol{H}_u^{(l+1)} = \text{UPDATE}^{(l)} \left( \boldsymbol{H}_u^{(l)}, \text{AGGREGATE}^{(l)} \left( \left\{ \boldsymbol{H}_v^{(l)}, \forall v \in \mathcal{N}(u) \right\} \right) \right), \quad (1)$$

where $\boldsymbol{H}^{(l+1)} \in \mathbb{R}^{n \times d}$ is the node features computed after $l$-steps of the GNN simplified as follows: $\boldsymbol{H}^{(l+1)} = \text{GNN}^{(l)}(\boldsymbol{H}^{(l)}, \boldsymbol{A}^{(l)})$, UPDATE and AGGREGATE are arbitrary differentiable functions, $\mathcal{N}(u)$ denotes a set of neighboring nodes of $u$, and $\boldsymbol{H}_u^{(1)}$ is initialized as the input node features $\boldsymbol{X}_u$.

**Graph Pooling** While message-passing functions can produce a set of node representations, we need an additional READOUT function to obtain an entire graph representation $\boldsymbol{h}_G \in \mathbb{R}^d$ as follows:

$$\boldsymbol{h}_G = \text{READOUT}\left(\{\boldsymbol{H}_v \mid v \in \mathcal{V}\}\right). \tag{2}$$

As a READOUT function, we can simply use the average or sum over all node features $\boldsymbol{H}_v, \forall v \in \mathcal{V}$ from the given graph (Atwood & Towsley, 2016; Xu et al., 2019). However, since such aggregation schemes take all node information equally without considering the graph structures, they lose structural information that is necessary for accurately representing a graph. To tackle this limitation, **Node Drop** methods (Gao & Ji, 2019; Lee et al., 2019b) select the high scored nodes $\boldsymbol{i}^{(l+1)} \in \mathbb{R}^{n_{l+1}}$ with learnable score function $s$ at layer $l$, to drop the unnecessary nodes, denoted as follows:

$$\boldsymbol{y}^{(l)} = s(\boldsymbol{H}^{(l)}, \boldsymbol{A}^{(l)}); \quad \boldsymbol{i}^{(l+1)} = \text{top}_k(\boldsymbol{y}^{(l)}), \tag{3}$$

where function $s$ depends on specific implementations, and $\text{top}_k$ function samples the top k nodes by dropping nodes with low scores $\boldsymbol{y}^{(l)} \in \mathbb{R}^{n_l}$. Whereas **Node Clustering** methods (Ying et al., 2018; Bianchi et al., 2019) learn a cluster assignment matrix $\boldsymbol{C}^{(l)} \in \mathbb{R}^{n_l \times n_{l+1}}$ with node features $\boldsymbol{H}^{(l)} \in \mathbb{R}^{n_l \times d}$, to coarsen the nodes and the adjacency matrix $\boldsymbol{A}^{(l)} \in \mathbb{R}^{n_l \times n_l}$ at layer $l$ as follows:

$$\boldsymbol{H}^{(l+1)} = \boldsymbol{C}^{(l)^T} \boldsymbol{H}^{(l)}; \quad \boldsymbol{A}^{(l+1)} = \boldsymbol{C}^{(l)^T} \boldsymbol{A}^{(l)} \boldsymbol{C}^{(l)}, \tag{4}$$

where generating an assignment matrix $\boldsymbol{C}^{(l)}$ depends on specific implementations. While these two approaches obtain decent performances on graph classification tasks, they are suboptimal since node drop methods unnecessarily drop arbitrary nodes, and node clustering methods have limited scalability to large graphs (Cangea et al., 2018; Lee et al., 2019b). Therefore, we need a sophisticated graph pooling layer that coarsens the graph with sparse implementation without discarding nodes.

## 3.2 GRAPH MULTISET TRANSFORMER

We now describe the *Graph Multiset Transformer* (GMT) architecture, which can accurately represent the entire graph, given a multiset of node features. We first introduce a multiset encoding scheme that allows to embed two different graphs into distinct embeddings, and then describe the graph multi-head attention that reflects the graph topology in the attention-based multiset encoding.

**Multiset Encoding** The input of the graph pooling function READOUT consists of nodes in a graph, and they form a multiset (i.e. a set that allows for repeating elements) since different nodes can have identical feature vectors. To design a graph pooling function that is as powerful as the WL test, it needs to satisfy the permutation invariance and injectiveness over the multiset, since two non-isomorphic graphs should be embedded differently through the injective function. While the simple sum pooling satisfies the injectiveness over a multiset (Xu et al., 2019), it may treat all node embeddings equally without consideration of their relevance to the task. To resolve this issue, we consider attention mechanism on the multiset pooling function to capture structural dependencies among nodes within a graph, in which we can provably enjoy the expressive power of the WL test.

**Graph Multi-head Attention** To overcome the inability of simple pooling methods (e.g. sum) on distinguishing important nodes, we use the attention mechanism as the main component in our pooling scheme. Assume that we have $n$ node vectors, and the input of the *attention function* (Att) consists of query $\boldsymbol{Q} \in \mathbb{R}^{n_q \times d_k}$, key $\boldsymbol{K} \in \mathbb{R}^{n \times d_k}$ and value $\boldsymbol{V} \in \mathbb{R}^{n \times d_v}$, where $n_q$ is the number of query vectors, $n$ is the number of input nodes, $d_k$ is the dimensionlity of the key vector, and $d_v$ is the dimensionality of the value vector. Then we compute the dot product of the query with all keys, to put more weights on the relevant values, namely nodes, as follows: $\text{Att}(\boldsymbol{Q}, \boldsymbol{K}, \boldsymbol{V}) = w(\boldsymbol{Q}\boldsymbol{K}^T)\boldsymbol{V}$, where $w$ is an activation function. Instead of computing a single attention, we can further use a multi-head attention (Vaswani et al., 2017), by linearly projecting the query $\boldsymbol{Q}$, key $\boldsymbol{K}$, and value $\boldsymbol{V}$ $h$ times respectively to yield $h$ different representation subspaces. The output of the *multi-head attention function* (MH) then can be denoted as follows:

$$\text{MH}(\boldsymbol{Q}, \boldsymbol{K}, \boldsymbol{V}) = [O_1, ..., O_h]\boldsymbol{W}^O; \quad O_i = \text{Att}(\boldsymbol{Q}\boldsymbol{W}_i^Q, \boldsymbol{K}\boldsymbol{W}_i^K, \boldsymbol{V}\boldsymbol{W}_i^V), \tag{5}$$

where the operations for $h$ parallel projections are parameter matrices $\boldsymbol{W}_i^Q \in \mathbb{R}^{d_k \times d_k}$, $\boldsymbol{W}_i^K \in \mathbb{R}^{d_k \times d_k}$, and $\boldsymbol{W}_i^V \in \mathbb{R}^{d_v \times d_v}$. Also, the output projection matrix is $\boldsymbol{W}^O \in \mathbb{R}^{hd_v \times d_{model}}$, where $d_{model}$ is the output dimensionality for the multi-head attention (MH) function.

While multi-head attention is superior to trivial pooling methods such as sum or mean as it considers global dependencies among nodes, the MH function suboptimally generates the key $\boldsymbol{K}$ and value

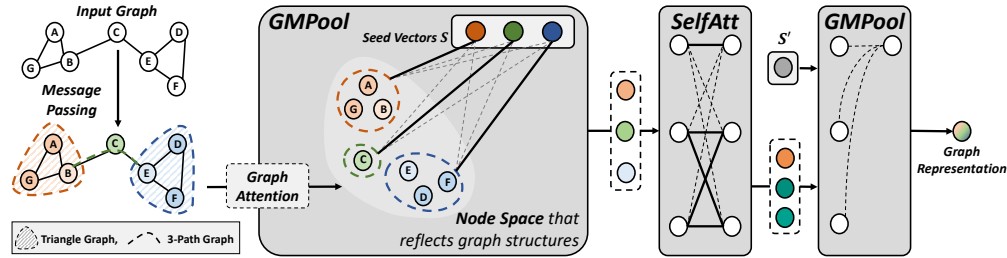

Figure 2: **Graph Multiset Transformer.** Given a graph passed through several message passing layers, we use an attention-based pooling block (GMPool) and a self-attention block (SelfAtt) to compress the nodes into few important nodes and consider the interaction among them respectively, within a multiset framework.

$\boldsymbol{V}$ for Att, since it linearly projects the obtained node embeddings $\boldsymbol{H}$ from equation 1 to further obtain the key and value pairs. To tackle this limitation, we newly define a novel *graph multi-head attention block* (GMH). Formally, given node features $\boldsymbol{H} \in \mathbb{R}^{n \times d}$ with their adjacency information $\boldsymbol{A}$, we construct the key and value using GNNs, to explicitly leverage the graph structure as follows:

$$\mathrm{GMH}(\boldsymbol{Q}, \boldsymbol{H}, \boldsymbol{A}) = [O_1, ..., O_h] \, \boldsymbol{W}^O; \quad O_i = \mathrm{Att}(\boldsymbol{Q}\boldsymbol{W}_i^Q, \mathrm{GNN}_i^K(\boldsymbol{H}, \boldsymbol{A}), \mathrm{GNN}_i^V(\boldsymbol{H}, \boldsymbol{A})), \quad (6)$$

where the output of $\mathrm{GNN}_i$ contains neighboring information of the graph, compared to the linearly projected node embeddings $\boldsymbol{K}\boldsymbol{W}_i^K$ and $\boldsymbol{V}\boldsymbol{W}_i^V$ in equation 5, for key and value matrices in Att.

**Graph Multiset Pooling with Graph Multi-head Attention**    Using the ingredients above, we now propose a graph pooling function that satisfies the injectiveness and permutation invariance, such that the overall architecture can be at most as powerful as the WL test, while taking the graph structure into account. Given node features $\boldsymbol{H} \in \mathbb{R}^{n \times d}$ from GNNs, we define a *Graph Multiset Pooling* (GMPool), which is inspired by the Transformer (Vaswani et al., 2017; Lee et al., 2019a), to compress the $n$ nodes into the $k$ typical nodes, with a parameterized seed matrix $\boldsymbol{S} \in \mathbb{R}^{k \times d}$ for the pooling operation that is directly optimized in an end-to-end fashion, as follows (Figure 2-GMPool):

$$\mathrm{GMPool}_k(\boldsymbol{H}, \boldsymbol{A}) = \mathrm{LN}(\boldsymbol{Z} + \mathrm{rFF}(\boldsymbol{Z})); \quad \boldsymbol{Z} = \mathrm{LN}(\boldsymbol{S} + \mathrm{GMH}(\boldsymbol{S}, \boldsymbol{H}, \boldsymbol{A})), \quad (7)$$

where rFF is any row-wise feedforward layer that processes each individual row independently and identically, and LN is a layer normalization (Ba et al., 2016). Note that the GMH function in equation 7 considers interactions between $k$ seed vectors (queries) in $\boldsymbol{S}$ and $n$ nodes (keys) in $\boldsymbol{H}$, to compress $n$ nodes into $k$ clusters with their attention similarities between queries and keys. Also, to extend the pooling scheme from set to multiset, we simply consider redundant node representations.

**Self-Attention for Inter-node Relationship**    While previously described GMPool condenses entire nodes into $k$ representative nodes, a major drawback of this scheme is that it does not consider relationships between nodes. To tackle this limitation, one should further consider the interactions among $n$ or condensed $k$ different nodes. To this end, we propose a *Self-Attention function* (SelfAtt), inspired by the Transformer (Vaswani et al., 2017; Lee et al., 2019a), as follows (Figure 2-SelfAtt):

$$\mathrm{SelfAtt}(\boldsymbol{H}) = \mathrm{LN}(\boldsymbol{Z} + \mathrm{rFF}(\boldsymbol{Z})); \quad \boldsymbol{Z} = \mathrm{LN}(\boldsymbol{H} + \mathrm{MH}(\boldsymbol{H}, \boldsymbol{H}, \boldsymbol{H})), \quad (8)$$

where, compared to GMH in equation 7 that considers interactions between $k$ vectors and $n$ nodes, SelfAtt captures inter-relationships among $n$ nodes by putting node embeddings $\boldsymbol{H}$ on both query and key locations in MH of equation 8. To satisfy the injectiveness property of SelfAtt, it might not consider interactions among $n$ nodes, which we discuss in Proposition 3 of Appendix A.1.

**Overall Architecture**    We now describe the full structure of *Graph Multiset Transformer* (GMT) consisting of GNN and pooling layers using ingredients above (See Figure 2). For a graph $G$ with node features $\boldsymbol{X}$ and an adjacency matrix $\boldsymbol{A}$, the Encoder : $G \mapsto \boldsymbol{H} \in \mathbb{R}^{n \times d}$ is denoted as follows:

$$\mathrm{Encoder}(\boldsymbol{X}, \boldsymbol{A}) = \mathrm{GNN}_2(\mathrm{GNN}_1(\boldsymbol{X}, \boldsymbol{A}), \boldsymbol{A}), \quad (9)$$

where we can stack several GNNs to construct the deep structures. After obtaining a set of node features $\boldsymbol{H}$ from an encoder, the pooling layer aggregates the features into a single vector form; Pooling : $\boldsymbol{H}, \boldsymbol{A} \mapsto \boldsymbol{h}_G \in \mathbb{R}^d$. To deal with a large number of nodes, we first condense the entire graph into $k$ representative nodes with *Graph Multiset Pooling* (GMPool), which is also adaptable to the varying size of nodes, and then utilize the interaction among them with *Self-Attention Block* (SelfAtt). Finally, we get the entire graph representation by using GMPool with $k = 1$ as follows:

$$\mathrm{Pooling}(\boldsymbol{H}, \boldsymbol{A}) = \mathrm{GMPool}_1(\mathrm{SelfAtt}(\mathrm{GMPool}_k(\boldsymbol{H}, \boldsymbol{A})), \boldsymbol{A}'), \quad (10)$$

where $\boldsymbol{A}' \in \mathbb{R}^{k \times k}$ is the identity or coarsened adjacency matrix since adjacency information should be adjusted after compressing the nodes from $n$ to $k$ with $\mathrm{GMPool}_k$ (See Appendix B for detail).

### 3.3 CONNECTION WITH WEISFEILER-LEHMAN GRAPH ISOMORPHISM TEST

Weisfeiler-Lehman (WL) test (Weisfeiler & Leman, 1968) is known for its ability to efficiently distinguish two different graphs. Recent studies (Morris et al., 2019; Xu et al., 2019) show that GNNs can be made to be as powerful as the WL test, by using an injective function over a multiset to map two different graphs into distinct spaces. Building on previous powerful GNNs, if our graph pooling function is injective, then our overall architecture can be at most as powerful as the WL test. To do so, we first recount the theorem from Xu et al. (2019), as formalized in Theorem 1.

**Theorem 1 (Non-isomorphic Graphs to Different Embeddings).** *Let $\mathcal{A} : \mathcal{G} \to \mathbb{R}^d$ be a GNN, and Weisfeiler-Lehman test decides two graphs $G_1 \in \mathcal{G}$ and $G_2 \in \mathcal{G}$ as non-isomorphic. Then, $\mathcal{A}$ maps two different graphs $G_1$ and $G_2$ to distinct vectors if node aggregation and update functions are injective, and graph-level readout, which operates on a multiset of node features $\{\boldsymbol{H}_i\}$, is injective.*

Since we focus on the representation of graphs through pooling, we deal with the injectiveness of the READOUT function. Our next Lemma 2 states that GMPool can represent the injective function.

**Lemma 2 (Injectiveness on Graph Multiset Pooling).** *Assume the input feature space $\mathcal{H}$ is a countable set. Then the output of $GMPool_k^i(\boldsymbol{H}, \boldsymbol{A})$ with $GMH(\boldsymbol{S}_i, \boldsymbol{H}, \boldsymbol{A})$ for a seed vector $\boldsymbol{S}_i$ can be unique for each multiset $\boldsymbol{H} \subset \mathcal{H}$ of bounded size. Further, the output of full $GMPool_k(\boldsymbol{H}, \boldsymbol{A})$ constructs a multiset with $k$ elements, which are also unique on the input multiset $\boldsymbol{H}$.*

**All proofs for the WL test are provided in Appendix A.1**. Based upon the injectiveness of GMPool, we show injectiveness of SelfAtt, to make an overall architecture (sequence of GMPool and SelfAtt with GNNs) as powerful as the WL test, formalized in Proposition 3. To satisfy the injectiveness of SelfAtt, we might not care about interactions among multiset elements (See Appendix A.1).

**Proposition 3 (Injectiveness on Pooling Function).** *The overall Graph Multiset Transformer with multiple GMPool and SelfAtt can map two different graphs $G_1$ and $G_2$ to distinct embedding spaces, such that the resulting GNN with proposed pooling functions can be as powerful as the WL test.*

### 3.4 CONNECTION WITH NODE CLUSTERING APPROACHES

Node clustering is widely used for coarsening a graph in a hierarchical manner, as described in the equation 4. However, since they require to store and even multiply the adjacency matrix $\boldsymbol{A}$ with the soft assignment matrix $\boldsymbol{C}$: $\boldsymbol{A}^{(l+1)} = \boldsymbol{C}^{(l)T} \boldsymbol{A}^{(l)} \boldsymbol{C}^{(l)}$, they need a quadratic space $\mathcal{O}(n^2)$ for $n$ nodes, which is problematic for large graphs. Meanwhile, our GMPool does not compute a coarsened adjacency matrix $A^{(l+1)}$, such that graph pooling is possible only with a sparse implementation, as formalized in Theorem 4. **All proofs regarding node clustering are provided in Appendix A.2.**

**Theorem 4 (Space Complexity of Graph Multiset Pooling).** *Graph Multiset Pooling condsense a graph with $n$ nodes to $k$ nodes in $\mathcal{O}(nk)$ space complexity, which can be further optimized to $\mathcal{O}(n)$.*

In spite of this huge strength on space complexity, our GMPool can be further approximated to the node clustering methods by manipulating an adjacency matrix, as formalized in Proposition 5.

**Proposition 5 (Approximation to Node Clustering).** *Graph Multiset Pooling $GMPool_k$ can perform hierarchical node clustering with learnable $k$ cluster centroids by Seed Vector $S$ in equation 7.*

Note that, contrary to previous node clusterings (Ying et al., 2018; Bianchi et al., 2019), GMPool learns data dependent $k$ cluster centroids that might be more meaningful to capture graph structures.

## 4 EXPERIMENT

To validate the proposed *Graph Multiset Transformer* (GMT) for graph representation learning, we evaluate it on classification, reconstruction and generation tasks of synthetic and real-world graphs.

### 4.1 GRAPH CLASSIFICATION

**Objective**    The goal of graph classification is to predict a label $\boldsymbol{y}_i \in \mathcal{Y}$ of a given graph $G_i \in \mathcal{G}$, with a mapping function $f : \mathcal{G} \to \mathcal{Y}$. To this end, we use a set of node representations $\{\boldsymbol{H}_v \mid v \in \mathcal{V}\}$ to obtain an entire graph representation $\boldsymbol{h}_G$ that is used to classify a label $f(G) = \hat{\boldsymbol{y}}$. We then learn $f$ with a cross-entropy loss, to minimize the negative log likelihood as follows: $\min \sum_{i=1} -\boldsymbol{y}_i \log \hat{\boldsymbol{y}}_i$.

Table 1: **Graph classification results** on test sets. The reported results are mean and standard deviation over 10 different runs. Best performance and its comparable results ($p > 0.05$) from the t-test are marked in bold. Hyphen (-) denotes out-of-resources that take more than 10 days (See Figure 4 for the time efficiency analysis).

| | Biochemical Domain | | | | | | | Social Domain | | | Significance |
|---|---|---|---|---|---|---|---|---|---|---|---|
| | D&D | PROTEINS | MUTAG | HIV | Tox21 | ToxCast | BBBP | IMDB-B | IMDB-M | COLLAB | |
| # graphs | 1,178 | 1,113 | 188 | 41,127 | 7,831 | 8,576 | 2,039 | 1,000 | 1,500 | 5,000 | - |
| # classes | 2 | 2 | 2 | 2 | 12 | 617 | 2 | 2 | 3 | 3 | - |
| Avg # nodes | 284.32 | 39.06 | 17.93 | 25.51 | 18.57 | 18.78 | 24.06 | 19.77 | 13.00 | 74.49 | - |
| GCN | 72.05 ± 0.55 | 73.24 ± 0.73 | 69.50 ± 1.78 | **76.81 ± 1.01** | 75.04 ± 0.80 | 60.63 ± 0.51 | 65.47 ± 1.73 | **73.26 ± 0.46** | 50.39 ± 0.41 | **80.59 ± 0.27** | 3 / 10 |
| GIN | 70.79 ± 1.17 | 71.46 ± 1.66 | 81.39 ± 1.53 | 75.95 ± 1.35 | 73.27 ± 0.84 | 60.83 ± 0.46 | **67.65 ± 3.00** | **72.78 ± 0.86** | 48.13 ± 1.36 | 78.19 ± 0.63 | 2 / 10 |
| Set2Set | 71.94 ± 0.56 | 73.27 ± 0.85 | 69.89 ± 1.94 | 74.70 ± 1.65 | 74.10 ± 1.13 | 59.70 ± 1.04 | 66.79 ± 1.05 | **72.90 ± 0.75** | 50.19 ± 0.39 | 79.55 ± 0.39 | 1 / 10 |
| SortPool | 75.58 ± 0.72 | 73.17 ± 0.88 | 71.94 ± 3.55 | 71.82 ± 1.63 | 69.54 ± 0.75 | 58.69 ± 1.71 | 65.98 ± 1.70 | 72.12 ± 1.12 | 48.18 ± 0.83 | 77.87 ± 0.47 | 0 / 10 |
| DiffPool | 77.56 ± 0.41 | 73.03 ± 1.00 | 79.22 ± 1.02 | 75.64 ± 1.86 | 74.88 ± 0.81 | 62.28 ± 0.56 | **68.25 ± 0.96** | **73.14 ± 0.70** | **51.31 ± 0.72** | 78.68 ± 0.43 | 3 / 10 |
| SAGPool(G) | 71.54 ± 0.91 | 72.02 ± 1.08 | 76.78 ± 2.12 | 74.56 ± 1.69 | 71.10 ± 1.06 | 59.88 ± 0.79 | 65.16 ± 1.93 | 72.16 ± 0.88 | 49.47 ± 0.56 | 78.85 ± 0.56 | 0 / 10 |
| SAGPool(H) | 74.72 ± 0.82 | 71.56 ± 1.49 | 73.67 ± 4.28 | 71.44 ± 1.67 | 69.81 ± 1.75 | 58.91 ± 0.80 | 63.94 ± 2.59 | **72.55 ± 1.28** | 50.23 ± 0.44 | 78.03 ± 0.31 | 1 / 10 |
| TopKPool | 73.63 ± 0.55 | 70.48 ± 1.01 | 67.61 ± 3.36 | 72.27 ± 0.91 | 69.39 ± 2.02 | 58.42 ± 0.91 | 65.19 ± 2.30 | 71.58 ± 0.95 | 48.59 ± 0.72 | 77.58 ± 0.85 | 0 / 10 |
| MinCutPool | **78.22 ± 0.54** | **74.72 ± 0.48** | 79.17 ± 1.64 | 75.37 ± 2.05 | 75.11 ± 0.69 | 62.48 ± 1.33 | 65.97 ± 1.13 | 72.65 ± 0.75 | **51.04 ± 0.70** | **80.87 ± 0.34** | 4 / 10 |
| StructPool | **78.45 ± 0.40** | **75.16 ± 0.86** | 79.50 ± 1.75 | 75.85 ± 1.81 | 75.43 ± 0.79 | 62.17 ± 1.61 | **67.01 ± 2.65** | 72.06 ± 0.64 | 50.23 ± 0.53 | 77.27 ± 0.51 | 3 / 10 |
| ASAP | 76.58 ± 1.04 | 73.92 ± 0.63 | 77.83± 1.49 | 72.86 ± 1.40 | 72.24 ± 1.66 | 58.09 ± 1.62 | 63.50 ± 2.47 | 72.81 ± 0.50 | **50.78 ± 0.75** | 78.64 ± 0.50 | 1 / 10 |
| EdgePool | 75.85 ± 0.58 | **75.12 ± 0.76** | 74.17± 1.82 | 72.66 ± 1.70 | 73.77 ± 0.68 | 60.70 ± 0.92 | **67.18 ± 1.97** | 72.46 ± 0.74 | **50.79 ± 0.59** | - | 3 / 9 |
| HaarPool | - | - | 66.11± 1.50 | - | - | - | 66.11 ± 0.82 | **73.29 ± 0.34** | 49.98 ± 0.57 | - | 1 / 5 |
| GMT (Ours) | **78.72 ± 0.59** | **75.09 ± 0.59** | **83.44 ± 1.33** | **77.56 ± 1.25** | **77.30 ± 0.59** | **65.44 ± 0.58** | **68.31 ± 1.62** | **73.48 ± 0.76** | **50.66 ± 0.82** | **80.74 ± 0.54** | **10 / 10** |

| Model | D&D | PROTEINS | BBBP |
|---|---|---|---|
| GMT | **78.72** | **75.09** | **68.31** |
| w/o message passing | 78.06 | 75.07 | 65.26 |
| w/o graph attention | 78.08 | 74.50 | 66.21 |
| w/o self-attention | 75.13 | 74.22 | 64.53 |
| mean pooling | 72.05 | 73.24 | 65.47 |

Table 2: Ablation Study of GMT on the D&D, PROTEINS, and BBBP datasets for graph classification.

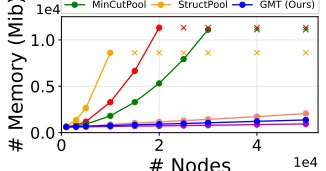

Figure 3: Memory efficiency of GMT compared with baselines. X indicates out-of-memory error.

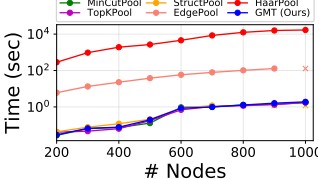

Figure 4: Time efficiency of GMT compared with baselines. X indicates out-of-memory error.

**Datasets** Among TU datasets (Morris et al., 2020), we select 6 datasets including 3 datasets (D&D, PROTEINS, and MUTAG) on Biochemical domain, and 3 datasets (IMDB-B, IMDB-M, and COL-LAB) on Social domain with accuracy for evaluation metric. Also, we use 4 molecule datasets (HIV, Tox21, ToxCast, BBBP) from the OGB datasets (Hu et al., 2020) with ROC-AUC for evaluation metric. Statistics are reported in the Table 1, and more details are described in the Appendix C.2.

**Models** **1) GCN. 2) GIN.** GNNs with mean or sum pooling (Kipf & Welling, 2017; Xu et al., 2019). **3) Set2Set.** Set pooling baseline (Vinyals et al., 2016). **4) SortPool. 5) SAGPool. 6) TopKPool. 7) ASAP.** The methods (Zhang et al., 2018; Lee et al., 2019b; Gao & Ji, 2019; Ranjan et al., 2020) that use the node drop, by dropping nodes (or clusters) with lower scores using scoring functions. **8) DiffPool. 9) MinCutPool. 10) HaarPool. 11) StructPool.** The methods (Ying et al., 2018; Bianchi et al., 2019; Wang et al., 2019; Yuan & Ji, 2020) that use the node clustering, by grouping a set of nodes into a set of clusters using a cluster assignment matrix. **12) EdgePool.** The method (Diehl, 2019) that gradually merges two adjacent nodes that have a high score edge. **13) GMT.** The proposed Graph Multiset Transformer (See Appendix C.1 for detailed descriptions).

**Implementation Details** For a fair comparison of pooling baselines (Lee et al., 2019b), we fix the GCN (Kipf & Welling, 2017) as a message passing layer. We evaluate the model performance on TU datasets for 10-fold cross validation (Zhang et al., 2018; Xu et al., 2019) with LIBSVM (Chang & Lin, 2011). Also, we use the initial node features following the fair comparison setup (Errica et al., 2020). We evaluate the performance on OGB datasets with their original feature extraction and data split settings (Hu et al., 2020). Experimental details are described in the Appendix C.2.

**Classification Results** Table 1 shows that our GMT outperforms most baselines, or achieves comparable performance to the best baseline results. These results demonstrate that our method is simple yet powerful as it only performs a single global operation at the final layer, unlike several baselines that use multiple pooling with a sequence of message passing (See Figure 9 for the detailed model architectures). Note that, since graph classification tasks mostly require the discriminative information to predict the labels of a graph, GNN baselines without parametric pooling, such as GCN and GIN, sometimes outperform pooling baselines on some datasets. In addition, recent work (Mesquita et al., 2020), which reveals that message-passing layers are dominant in the graph classification, supports this phenomenon. Therefore, we conduct experiments on graph reconstruction to directly quantify the amount of retained information after pooling, which we describe in the next subsection.

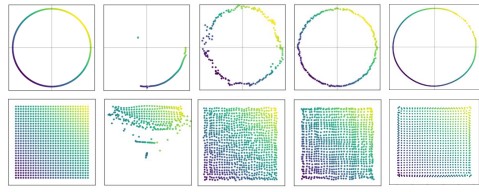

(a) Original  (b) TopKPool  (c) DiffPool  (d) MinCutPool  (e) GMPool

Figure 5: Reconstruction results of ring and grid synthetic graphs, compared to node drop and clustering methods. See Figure 10 for high resolution.

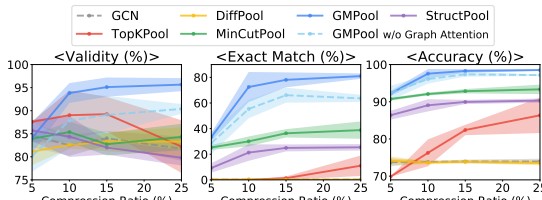

Figure 6: Reconstruction results on the ZINC molecule dataset by varying the compression ratio. Solid lines denote the mean, and shaded areas denote the variance.

**Ablation Study**   To see where the performance improvement comes from, we conduct an ablation study on GMT by removing graph attention, self attention, and message-passing operations. Table 2 shows that using graph attention with self-attention helps significantly improve the performances from the mean pooling. Further, performances of the GMT without message-passing layers indicate that our pooling layer well captures the graph multiset structure only with pooling without GNNs.

**Efficiency**   While node clustering methods achieve decent performances in Table 1, they are known to suffer from large memory usage since they cannot work with sparse graph implementations. To compare the GPU **Memory Efficiency** of GMT with baseline models, we test it on the Erdos-Renyi graphs (Erdős & Rényi, 1960) (See Appendix C.2 for detail setup). Figure 3 shows that our GMT is highly efficient in terms of memory thanks to its compatibility with sparse graphs, making it more practical over memory-heavy pooling baselines. In addition to this, we measure the **Time Efficiency** to further validate the practicality of GMT in terms of time complexity. We validate it with the same Erdos-Renyi graphs (See Appendix C.2 for detail setup). Figure 4 shows that GMT takes less than (or nearly about) a second even for large graphs, compared to the slowly working models such as HaarPool and EdgePool. This result further confirms that our GMT is practically efficient.

## 4.2   GRAPH RECONSTRUCTION

Graph classification does not directly measure the expressiveness of GNNs since identifying discriminative features may be more important than accurately representing graphs. Meanwhile, graph reconstruction directly quantifies the graph information retained by condensed nodes after pooling.

**Objective**   For graph reconstruction, we train an autoencoder to reconstruct the input node features $X \in \mathbb{R}^{n \times c}$ from their pooled representations $X^{pool} \in \mathbb{R}^{k \times c}$. The learning objective to minimize the discrepancy between the original graph $X$ and the reconstructed graph $X^{rec}$ with a cluster assignment matrix $C \in \mathbb{R}^{n \times k}$ is denoted as follows: $\min \|X - X^{rec}\|$, where $X^{rec} = CX^{pool}$.

**Experimental Setup**   We first experiment with **Synthetic Graph**, such as ring and grid (Bianchi et al., 2019), that can be represented in a 2-D Euclidean space, where the goal is to restore the location of each node from pooled features, with an adjacency matrix. We further experiment with real-world **Molecule Graph**, namely ZINC datasets (Irwin et al., 2012), which consists of 12K molecular graphs. See Appendix C.3 for the experimental details including model descriptions.

**Reconstruction Results**   Figure 5 shows the original and the reconstructed graphs for **Synthetic Graph** of ring and grid structures. The noisy results of baselines indicate that the condensed node features do not fully capture the original graph structure.   Whereas our GMPool yields almost perfect reconstruction, which demonstrates that our pooling operation learns meaningful representation without discarding the original graph information. We further validate the reconstruction performance of the proposed GMPool on the real-world **Molecule Graph**, namely ZINC, by varying the compression

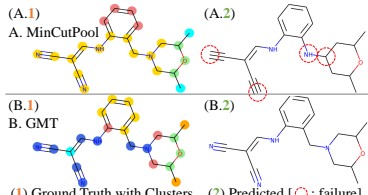

Figure 7: Reconstruction example with assigned clusters as colors on left and reconstructed molecules on right.

ratio. Figure 6 shows reconstruction results on the molecule graph, on which GMPool largely outperforms all compared baselines in terms of validity, exact match, and accuracy (High score indicates the better, and see Appendix C.3 for the detailed description of evaluation metrics). With given results, we demonstrate that our GMPool can be easily extended to the node clustering schemes, while it is powerful enough to encode meaningful information to reconstruct the graph.

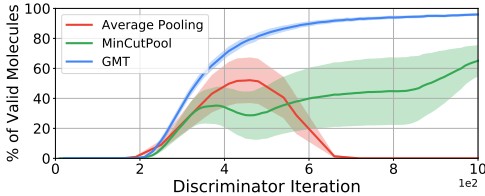

| Top-$k$ accuracy: | | 1 | 3 | 5 | 10 | 20 | 50 |
|---|---|---|---|---|---|---|---|
| Reaction | GLN | 51.41 | 67.55 | 74.92 | 83.48 | 88.64 | 92.37 |
| Class | MinCutPool | 51.17 | 67.47 | **75.59** | **83.68** | 89.31 | 92.31 |
| Unknown | GMT (Ours) | **51.83** | **68.20** | 75.17 | 83.20 | **89.33** | **92.47** |
| Reaction | GLN | 63.53 | 78.27 | 84.32 | 89.51 | 92.17 | 93.17 |
| Class | MinCutPool | 63.91 | 79.19 | 84.76 | 89.69 | 92.13 | 93.23 |
| as Prior | GMT (Ours) | **64.17** | **79.61** | **85.32** | **89.97** | **92.31** | **93.25** |

Figure 8: Validity curve for molecule generation on QM9 dataset from MolGAN. Solid lines denote the mean and shaded areas denote the variance.

Table 3: Top-k accuracy for Retrosynthesis experiment on USPTO-50k data, for cases where the reaction class is given as prior information (Bottom) and not given (Top).

**Qualitative Analysis**   We visualize the reconstruction examples from ZINC in Figure 7, where colors in the left figure indicate the assigned clusters on each atoms, and red dashed circles indicate the incorrectly predicted atoms on the reconstructed molecule. As shown in Figure 7, GMPool yields more calibrated clustering than MinCutPool, capturing the detailed substructures, which results in the successful reconstruction (See Figure 11 in Appendix D for more reconstruction examples).

### 4.3   GRAPH GENERATION

**Objective**   Graph generation is used to generate a valid graph that satisfies the desired properties, in which graph encoding is used to improve the generation performances. Formally, given a graph $G$ with graph encoding function $f$, the goal here is to generate a valid graph $\bar{G} \in \mathcal{G}$ of desired property $\boldsymbol{y}$ with graph decoding function $g$ as follows: $\min d(\boldsymbol{y}, \bar{G})$, where $\bar{G} = g(f(G))$. $d$ is a distance metric between the generated graph and desired properties, to guarantee that the graph has them.

**Experimental Setup**   To evaluate the applicability of our model, we experiment on **Molecule Generation** to stably generate the valid molecules with MolGAN (Cao & Kipf, 2018), and **Retrosynthesis** to empower the synthesis performances with Graph Logic Network (GLN) (Dai et al., 2019), by replacing their graph embedding function $f(G)$ to ours. In both experiments, we replace the average pooling to either the MinCutPool or GMT. See Appendix C.4 for more experimental details.

**Generation Results**   The power of a discriminator distinguishing whether a molecule is real or fake is highly important to create a valid molecule in MolGAN. Figure 8 shows the validity curve on the early stage of MolGAN training for **Molecule Generation**, and the representation power of GMT significantly leads to the stabilized generation of valid molecules than baselines. Further, Table 3 shows **Retrosynthesis** results, where we use the GLN as a backbone architecture. Similar to the molecule generation, retrosynthesis with GMT further improves the performances, which suggests that GMT can replace existing pooling methods for improved performances on diverse graph tasks.

## 5   CONCLUSION

In this work, we pointed out that existing graph pooling approaches either do not consider the task relevance of each node (sum or mean) or may not satisfy the injectiveness (node drop and clustering methods). To overcome such limitations, we proposed a novel graph pooling method, *Graph Multiset Transformer* (GMT), which not only encodes the given set of node embeddings as a multiset to uniquely embed two different graphs into two distinct embeddings, but also considers both the global structure of the graph and their task relevance in compressing the node features. We theoretically justified that the proposed pooling function is as powerful as the WL test, and can be extended to the node clustering schemes. We validated the proposed GMT on 10 graph classification datasets, and our method outperformed state-of-the-art graph pooling models on most of them. We further showed that our method is superior to the existing graph pooling approaches on graph reconstruction and generation tasks, which require more accurate representations of the graph than classification tasks. We strongly believe that the proposed pooling method will bring substantial practical impact, as it is generally applicable to many graph-learning tasks that are becoming increasingly important.

ACKNOWLEDGMENTS

This work was supported by Institute of Information & communications Technology Planning & Evaluation (IITP) grant funded by the Korea government (MSIT) (No.2019-0-00075, Artificial Intelligence Graduate School Program (KAIST)), and the National Research Foundation of Korea (NRF) grant funded by the Korea government (MSIT) (NRF-2018R1A5A1059921).

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

# A  PROOFS

## A.1  PROOFS REGARDING WEISFEILER-LEHMAN TEST

We first recount the theorem of Xu et al. (2019) to define a GNN that is injective over a multiset (Theorem 1). We then prove that the proposed *Graph Multiset Pooling* (GMPool) can map two different graphs to distinct spaces (Lemma 2). Finally, we show that our overall architecture *Graph Multiset Transformer* (GMT) with a sequence of proposed *Graph Multiset Pooling* (GMPool) and *Self-Attention* (SelfAtt) can represent the injective function over the input multiset (Proposition 3).

**Theorem 1 (Non-isomorphic Graphs to Different Embeddings).** *Let $\mathcal{A} : \mathcal{G} \to \mathbb{R}^d$ be a GNN, and Weisfeiler-Lehman test decides two graphs $G_1 \in \mathcal{G}$ and $G_2 \in \mathcal{G}$ as non-isomorphic. Then, $\mathcal{A}$ maps two different graphs $G_1$ and $G_2$ to distinct vectors if node aggregation and update functions are injective, and graph-level readout, which operates on a multiset of node features $\{\boldsymbol{H}_i\}$, is injective.*

*Proof.* To map two non-isomorphic graphs to distinct embedding spaces with GNNs, we recount the theorem on Graph Isomorphism Network. See Appendix B of Xu et al. (2019) for details.  $\square$

**Lemma 2 (Uniqueness on Graph Multiset Pooling).** *Assume the input feature space $\mathcal{H}$ is a countable set. Then the output of the $GMPool_k^i(\boldsymbol{H}, \boldsymbol{A})$ with $GMH(\boldsymbol{S}_i, \boldsymbol{H}, \boldsymbol{A})$ for a seed vector $\boldsymbol{S}_i$ can be unique for each multiset $\boldsymbol{H} \subset \mathcal{H}$ of bounded size. Further, the output of the full $GMPool_k(\boldsymbol{H}, \boldsymbol{A})$ constructs a multiset with k elements, which are also unique on the input multiset $\boldsymbol{H}$.*

*Proof.* We first state that the GNNs of the Graph Multi-head Attention (GMH) in a GMPool can represent the injective function over the multiset $\boldsymbol{H}$ with an adjacency information $\boldsymbol{A}$, by selecting proper message-passing functions that satisfy the WL test (Xu et al., 2019; Morris et al., 2019), denoted as follows: $\boldsymbol{H}' = \mathrm{GNN}(\boldsymbol{H}, \boldsymbol{A})$, where $\boldsymbol{H}' \subset \mathcal{H}$. Then, given enough elements, a $\mathrm{GMPool}_k^i(\boldsymbol{H}, \boldsymbol{A})$ can express the sum pooling over the multiset $\boldsymbol{H}'$ defined as follows: $\rho(\sum_{\boldsymbol{h} \in \boldsymbol{H}'} f(\boldsymbol{h}))$, where $f$ and $\rho$ are mapping functions (see the proof of PMA in Lee et al. (2019a)).

Since $\mathcal{H}$ is a countable set, there is a mapping from the elements to prime numbers denoted by $p(\boldsymbol{h}) : \mathcal{H} \to \mathbb{P}$. If we let $f(\boldsymbol{h}) = -\log p(\boldsymbol{h})$, then $\sum_{\boldsymbol{h} \in \boldsymbol{H}'} f(\boldsymbol{h}) = \log \prod_{\boldsymbol{h} \in \boldsymbol{H}'} \frac{1}{p(\boldsymbol{h})}$ which constitutes an unique mapping for every multiset $\boldsymbol{H}' \subset \mathcal{H}$ (see Wagstaff et al. (2019)). In other words, $\sum_{\boldsymbol{h} \in \boldsymbol{H}'} f(\boldsymbol{h})$ is injective. Also, we can easily construct a function $\rho$, such that $\mathrm{GMPool}_k^i(\boldsymbol{H}, \boldsymbol{A}) = \rho(\sum_{\boldsymbol{h} \in \boldsymbol{H}'} f(\boldsymbol{h})) = \rho(\log \prod_{\boldsymbol{h} \in \boldsymbol{H}'} \frac{1}{p(\boldsymbol{h})})$ is the injective function for every multiset $\boldsymbol{H} \subset \mathcal{H}$, where $\boldsymbol{H}'$ is derived from the GNN component in the GMPool; $\boldsymbol{H}' = \mathrm{GNN}(\boldsymbol{H}, \boldsymbol{A})$.

Furthermore, since a GMPool considers multiset elements without any order, it satisfies the permutation invariance condition for the multiset function.

Finally, each GMPool block has $k$ components such that the output of it consists of $k$ elements as follows: $\mathrm{GMPool} = \left\{\mathrm{GMPool}_k^i(\boldsymbol{H}, \boldsymbol{A})\right\}_{i=1}^{k}$, which allows multiple instances for its elements. Then, since each $\mathrm{GMPool}_k^i(\boldsymbol{H}, \boldsymbol{A})$ is unique on the input multiset $\boldsymbol{H}$, the output of the GMPool that consists of $k$ outputs is also unique on the input multiset $\boldsymbol{H}$.  $\square$

Thanks to the universal approximation theorem (Hornik et al., 1989), we can construct such functions $p$ and $\rho$ using multi-layer perceptrons (MLPs).

**Proposition 3 (Injectiveness on Pooling Function).** *The overall Graph Multiset Transformer with multiple GMPool and SelfAtt can map two different graphs $G_1$ and $G_2$ to distinct embedding spaces, such that the resulting GNN with proposed pooling functions can be as powerful as the WL test.*

*Proof.* By Lemma 2, we know that a *Graph Multiset Pooling* (GMPool) can represent the injective function over the input multiset $\boldsymbol{H} \subset \mathcal{H}$. If we can also show that a *Self-Attention* (SelfAtt) can represent the injective function over the multiset, then the sequence of the GMPool and SelfAtt blocks can satisfy the injectiveness.

Let $\boldsymbol{W}^O$ be a zero matrix in SelfAtt function. SelfAtt($\boldsymbol{H}$) then can be approximated to the any instance-wise feed-forward network denoted as follows: SelfAtt($\boldsymbol{H}$) = rFF($\boldsymbol{H}$). Therefore, this

rFF is a suitable transformation $\phi : \mathbb{R}^d \rightarrow \mathbb{R}^d$ that can be easily constructed over the multiset elements $\boldsymbol{h} \in \boldsymbol{H}$, to satisfy the injectiveness. □

To maximize the discriminative power of the Graph Multiset Transformer (GMT) by satisfying the WL test, we assume that SelfAtt does not consider the interactions among multiset elements, namely nodes. While proper GNNs with the proposed pooling function can be at most as powerful as the WL test with this assumption, our experimental results with the ablation study show that the interaction among nodes is significantly important to distinguish the broad classes of graphs (See Table 2).

### A.2 PROOFS REGRADING NODE CLUSTERING

We first prove that the space complexity of the *Graph Multiset Pooling* (GMPool) without GNNs can be approximated to the $\mathcal{O}(n)$ with $n$ nodes (Theorem 4). After that, we show that the GMPool can be extended to the node clustering approaches with learnable cluster centroids (Proposition 5).

**Theorem 4 (Space Complexity of Graph Multiset Pooling).** *Graph Multiset Pooling condsense a graph with $n$ nodes to $k$ nodes in $\mathcal{O}(nk)$ space complexity, which can be further optimized to $\mathcal{O}(n)$.*

*Proof.* Assume that we have key $\boldsymbol{K} \in \mathbb{R}^{n \times d_k}$ and value $\boldsymbol{V} \in \mathbb{R}^{n \times d_v}$ matrices in the Att function of Graph Multi-head Attention (GMH) for the simplicity, which is described in the equation 6. Also, $\boldsymbol{Q}$ is defined as a seed vector $\boldsymbol{S} \in \mathbb{R}^{k \times d}$ in the GMPool function of the equation 7. To obtain the weights on the values $\boldsymbol{V}$, we multiply the query $\boldsymbol{Q}$ with key $\boldsymbol{K}$: $\boldsymbol{Q}\boldsymbol{K}^T$. This matrix multiplication then maps a set of $n$ nodes into a set of $k$ nodes, such that it requires $\mathcal{O}(nk)$ space complexity. Also, we can further drop the constant term $k$: $\mathcal{O}(n)$, by properly setting the small $k$ values; $k \ll n$.

The multiplication of the attention weights $\boldsymbol{Q}\boldsymbol{K}^T$ with value $\boldsymbol{V}$ also takes the same complexity, such that the overall space complexity of GMPool is $\mathcal{O}(nk)$, which can be further optimized to $\mathcal{O}(n)$. □

The space complexity of GNNs with sparse implementation requires $\mathcal{O}(n + m)$ space complexity, where $n$ is the number of nodes, and $m$ is the number of edges in a graph. Therefore, multiple GNNs followed by our GMPool require the total space complexity of $\mathcal{O}(n + m)$ due to the space complexity of the GNN operations. However, GNNs with our GMPool are more efficient than node clustering methods, since node clustering approaches need $\mathcal{O}(n^2)$ space complexity.

**Proposition 5 (Approximation to Node Clustering).** *Graph Multiset Pooling $GMPool_k$ can perform hierarchical node clustering with learnable $k$ cluster centroids by Seed Vector $S$ in equation 7.*

*Proof.* Node clustering approaches are widely used to coarsen a given large graph in a hierarchical manner with several message-passing functions. The core part of the node clustering schemes is to generate a cluster assignment matrix $\boldsymbol{C}$, to coarsen nodes and adjacency matrix as in an equation 4. Therefore, if our Graph Multiset Pooling (GMPool) can generate a cluster assignment matrix $\boldsymbol{C}$, then the proposed GMPool can be directly approximated to the node clustering approaches.

In the proposed GMPool, query $\boldsymbol{Q}$ is generated from a learnable set of $k$ seed vectors $\boldsymbol{S}$, and key $\boldsymbol{K}$ and value $\boldsymbol{V}$ are generated from node features $\boldsymbol{H}$ with GNNs in the Graph Multi-head Attention (GMH) block, as in an equation 6. In this function, if we decompose the attention function $\text{Att}(\boldsymbol{Q}, \boldsymbol{K}, \boldsymbol{V}) = w(\boldsymbol{Q}\boldsymbol{K}^T)\boldsymbol{V}$ into the dot products of the query with all keys, and the corresponding weighted sum of values, then the first dot product term inherently generates a soft assignment matrix as follows: $\boldsymbol{C} = w(\boldsymbol{Q}\boldsymbol{K}^T)$. Therefore, the proposed GMPool can be easily extended to the node clustering schemes, with the inherently generated cluster assignment matrix; $\boldsymbol{C} = w(\boldsymbol{Q}\boldsymbol{K}^T)$, where one of the proper choices for the activation function $w$ is the softmax function as follows:

$$w(\boldsymbol{Q}\boldsymbol{K}^T)_{i,j} = \frac{\exp(\boldsymbol{Q}_i \boldsymbol{K}_j^T)}{\sum_{n=1}^{k} \exp(\boldsymbol{Q}_n \boldsymbol{K}_j^T)}. \tag{11}$$

Furthermore, through the learnable seed vectors $\boldsymbol{S}$ for the query $\boldsymbol{Q}$, we can learn data dependent $k$ different cluster centroids in an end-to-end fashion. □

Note that, as shown in the section 4.2 of the main paper, the proposed GMPool significantly outperforms the previous node clustering approaches (Ying et al., 2018; Bianchi et al., 2019). This

is because, contrary to them, the proposed GMPool can explicitly learn data dependent $k$ cluster centroids by learnable seed vectors $S$.

# B   DETAILS FOR GRAPH MULTISET TRANSFORMER COMPONENTS

In this section, we describe the *Graph Multiset Pooling* (GMPool) and *Self-Attention* (SelfAtt), which are the components of the proposed *Graph Multiset Transformer*, in detail.

**Graph Multiset Pooling**   The core components of the Graph Multiset Pooling (GMPool) is the Graph Multi-head Attention (GMH) that considers the graph structure into account, by constructing the key $K$ and value $V$ using GNNs, as described in the equation 6. As shown in the Table 2 of the main paper, this graph multi-head attention significantly outperforms the naive multi-head attention (MH in equation 5). However, after compressing the $n$ nodes into the $k$ nodes with $\text{GMPool}_k$, we can not directly perform further GNNs since the original adjacency information is useless after pooling. To tackle this limitation, we can generate the new adjacency matrix $A'$ for the compressed nodes, by performing node clustering as described in Proposition 5 of the main paper as follows:

$$\text{GMPool}_1(\text{GMPool}_k(H, A), A'); \quad A' = C^T A C, \tag{12}$$

where $C$ is the generated cluster assignment matrix, and $A'$ is the coarsened adjacency matrix as described in the equation 4. However, this approach is well known for their scalability issues (Lee et al., 2019b; Cangea et al., 2018), since they require quadratic space $\mathcal{O}(n^2)$ to store and even multiply the adjacency matrix $A$ with the soft assignment matrix $C$. Therefore, we leave doing this as a future work, and use the following trick. By replacing the adjacency matrix $A$ with the identity matrix $I$ in the GMPool except for the first block, we can easily perform multiple GMPools without any constraints, which is approximated to the GMPool with MH in the equation 5, rather than GMH in the equation 6, as follows:

$$\text{GMPool}_1(\text{GMPool}_k(H, A), A'); \quad A' = I. \tag{13}$$

**Self-Attention**   The Self-Attention (SelfAtt) function can consider the inter-relationships between nodes in a set, which helps the network to take the global graph structure into account. Because of this advantage, the self-attention function significantly improves the proposed model performance on the graph classification tasks, as shown in the Table 2 of the main paper. From a different perspective, we can regard the Self-Attention function as a graph neural network (GNN) with a complete graph. Specifically, given $k$ nodes from the previous layer, the Multi-head Attention (MH) of the Self-Attention function first constructs the adjacency matrix among all nodes with their similarities, through the matrix multiplication of the query with key: $QK^T$, and then computes the outputs with the sum of the obtained weights on the value. In other words, the self-attention function can be considered as one message passing function with a soft adjacency matrix, which might be further connected to the Graph Attention Network (Velickovic et al., 2018).

# C   EXPERIMENTAL SETUP

In this section, we first introduce the baselines and our model, and then describe the experimental details about graph classification, reconstruction, and generation tasks respectively.

## C.1   BASELINES AND OUR MODEL

**1) GCN.** This method (Kipf & Welling, 2017) is the mean pooling baseline with Graph Convolutional Network (GCN) as a message passing layer.

**2) GIN.** This method (Xu et al., 2019) is the sum pooling baseline with Graph Isomorphism Network (GIN) as a message passing layer.

**3) Set2Set.** This method (Vinyals et al., 2016) is the set pooling baseline that uses a recurrent neural network to encode a set of all nodes, with content-based attention over them.

**4) SortPool.** This method (Zhang et al., 2018) is the node drop baseline that drops unimportant nodes by sorting their representations, which are directly generated from the previous GNN layers.

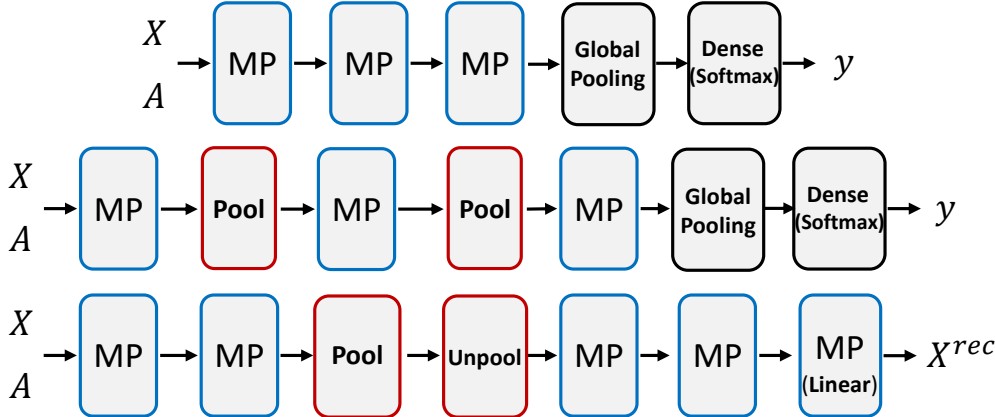

Figure 9: **Illustration of High-level Model Architectures. (Top):** Global Graph Classification; GCN, GIN, Set2Set, SortPool, SAGPool(G), StructPool, GMT. **(Middle:)** Hierarchical Graph Classification; DiffPool, SAGPool(H), TopKPool, MinCutPool, ASAP, EdgePool, HaarPool. **(Bottom:)** Graph Reconstruction; Diff-Pool, TopKPool, MinCutPool, GMT. MP denotes the message passing layer.

**5) SAGPool.** This method (Lee et al., 2019b) is the node drop baseline that selects the important nodes, by dropping unimportant nodes with lower scores that are generated by the another graph convolutional layer, instead of using scores from the previously passed layers. Particularly, this method has two variants. **6.1) SAGPool(G)** is the global node drop method that drops unimportant nodes one time at the end of their architecture. **6.2) SAGPool(H)** is the hierarchical node drop method that drops unimportant nodes sequentially with multiple graph convolutional layers.

**6) TopkPool.** This method (Gao & Ji, 2019) is the node drop baseline that selects the top-ranked nodes using a learnable scoring function.

**7) ASAP.** This method (Ranjan et al., 2020) is the node drop baseline that first locally generates the clusters with neighboring nodes, and then drops the lower score clusters using a scoring function.

**8) DiffPool.** This method (Ying et al., 2018) is the node clustering baseline that produces the hierarchical representation of the graphs in an end-to-end fashion, by clustering similar nodes into the few nodes through graph convolutional layers.

**9) MinCutPool.** This method (Bianchi et al., 2019) is the node clustering baseline that applies the spectral clustering with GNNs, to coarsen the nodes and the adjacency matrix of a graph.

**10) HaarPool.** This method (Wang et al., 2019) is the spectral-based pooling baseline that compresses the node features with a nonlinear transformation in a Haar wavelet domain. Since it directly uses the spectral clustering to generate a coarsened matrix, the time complexity cost is relatively higher than other pooling methods.

**11) StructPool.** This method (Yuan & Ji, 2020) is the node clustering baseline that integrates the concept of the conditional random field into the graph pooling. While this method can be used with a hierarchical scheme, we use it with a global scheme following their original implementation, which is similar to the SortPool (Zhang et al., 2018).

**12) EdgePool.** This method (Diehl, 2019) is the edge clustering baseline that gradually merges the nodes, by contracting the high score edge between two adjacent nodes.

**13) GMT.** Our Graph Multiset Transformer that first condenses all nodes into the important nodes by GMPool, and then considers interactions between nodes in a set. Since it operates on the global READOUT layer, it can be coupled with hierarchical pooling methods by replacing their last layer.

## C.2 GRAPH CLASSIFICATION

**Dataset**   Among TU datasets (Morris et al., 2020), we select the 6 datasets including 3 datasets (D&D, PROTEINS, and MUTAG) on Biochemical domain, and 3 datasets (IMDB-B, IMDB-M, and COLLAB) on Social domain. We use the classification accuracy as an evaluation metric. As suggested by Errica et al. (2020) for a fair comparison, we use the one-hot encoding of their atom

types as initial node features in the bio-chemical datasets, and the one-hot encoding of node degrees as initial node features in the social datasets. Moreover, we use the recently suggested 4 molecule graphs (HIV, Tox21, ToxCast, BBBP) from the OGB datasets (Hu et al., 2020). We use the ROC-AUC for an evaluation metric, and use the additional atom and bond features, as suggested by Hu et al. (2020). Dataset statistics are reported in the Table 1 of the main paper.

**Implementation Details on Classification Experiments**   For all experiments on TU datasets, we evaluate the model performance with a 10-fold cross validation setting, where the dataset split is based on the conventionally used training/test splits (Niepert et al., 2016; Zhang et al., 2018; Xu et al., 2019), with LIBSVM (Chang & Lin, 2011). In addition, we use the 10 percent of the training data as a validation data following the fair comparison setup (Errica et al., 2020). For all experiments on OGB datasets, we evaluate the model performance following the original training/validation/test dataset splits (Hu et al., 2020). We use the early stopping criterion, where we stop the training if there is no further improvement on the validation loss during 50 epochs, for the TU datasets. Further, the maximum number of epochs is set to 500. We then report the average performances on the validation and test sets, by performing overall experiments 10 times with different seeds.

For all experiments on TU datasets except the D&D, the learning rate is set to $5 \times 10^{-4}$, hidden size is set to 128, batch size is set to 128, weight decay is set to $1 \times 10^{-4}$, and dropout rate is set to 0.5. Since the D&D dataset has a large number of nodes (See Table 1 in the main paper), node clustering methods can not perform clustering operations on large graphs with large batch sizes, such that the hidden size is set to 32, and batch size is set to 10 on the D&D dataset. For all experiments on OGB datasets except the HIV, the learning rate is set to $1 \times 10^{-3}$, hidden size is set to 128, batch size is set to 128, weight decay is set to $1 \times 10^{-4}$, and dropout rate is set to 0.5. Since the HIV dataset contains a large number of graphs compared to others (See Table 1 in the main paper), the batch size is set to 512 for fast training. Then we optimize the network with Adam optimizer (Kingma & Ba, 2014). For a fair comparison of baselines (Lee et al., 2019b), we use the three GCN layers (Kipf & Welling, 2017) as a message passing function for all models with jumping knowledge strategies (Xu et al., 2018), and only change the pooling architecture throughout all models, as illustrated in Figure 9. Also, we set the pooling ratio as 25% in each pooling layer for both baselines and our models.

**Implementation Details on Efficiency Experiments**   To compare the GPU **memory efficiency** of GMT against baseline models including node drop and node clustering methods, we first generate the Erdos-Renyi graphs (Erdős & Rényi, 1960) by varying the number of nodes $n$, where the edge size $m$ is twice the number of nodes: $m = 2n$. For all models, we compress the given $n$ nodes into the $k = 4$ nodes at the first pooling function.

To compare the **time efficiency** of GMT against baseline models, we first generate the Erdos-Renyi graphs (Erdős & Rényi, 1960) by varying the number of nodes $n$ with $m = n^2/10$ edges, following the setting of HaarPool (Wang et al., 2019). For all models, we set the pooling ratio as 25% except for HaarPool, since it compresses the nodes according to the coarse-grained chain of a graph. We measure the forward time, including CPU and GPU, for all models with 50 graphs over one batch.

## C.3   GRAPH RECONSTRUCTION

**Dataset**   We first experiment with synthetic graphs represented in a 2-D Euclidean space, such as ring and grid structures. The node features of a graph consist of their location in a 2-D coordinate space, and the adjacency matrix indicates the connectivity pattern of nodes. The goal here is to restore all node locations from compressed features after pooling, with the intact adjacency matrix.

While synthetic graphs are appropriate choices for the qualitative analysis, we further do the quantitative evaluation of models with real-world molecular graphs. Specifically, we use the subset (Dwivedi et al., 2020) of the ZINC dataset (Irwin et al., 2012), which consists of 12K real-world molecular graphs, to further conduct a graph reconstruction on the large number of various graphs. The goal of the molecule reconstruction task is to restore the exact atom types of all nodes in the given graph, from the compressed representations after pooling.

**Common Implementation Details**   Following Bianchi et al. (2019), we use the two message passing layers both right before the pooling operation and right after the unpooling operation. Also, both pooling and unpooling operations are performed once and sequentially connected, as illustrated in

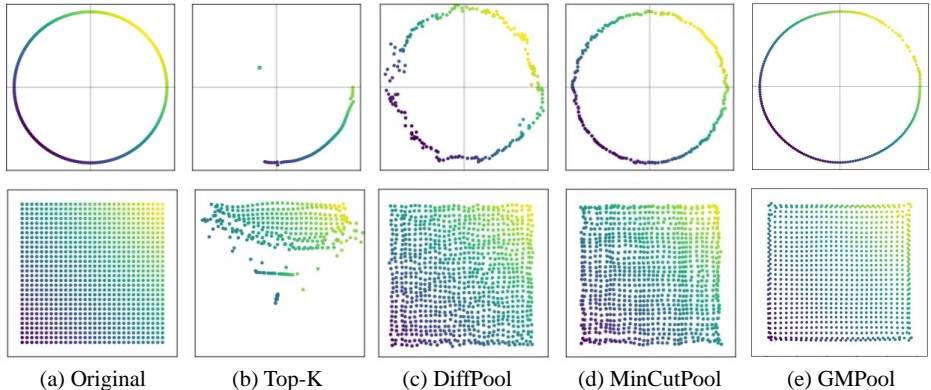

|  |  |  |  |  |
|:---:|:---:|:---:|:---:|:---:|
| (a) Original | (b) Top-K | (c) DiffPool | (d) MinCutPool | (e) GMPool |

Figure 10: High resolution images for synthetic graph reconstruction results in Figure 5.

the Figure 9. We compare our methods against both the node drop (TopKPool (Gao & Ji, 2019)) and node clustering (DiffPool (Ying et al., 2018) and MinCutPool (Bianchi et al., 2019)) methods. For the node drop method, we use the unpooling operation proposed in the graph U-net (Gao & Ji, 2019). For the node clustering methods, we use the graph coarsening schemes described in the equation 4, with their specific implementations on generating an assignment matrix. For our proposed method, we only use the one Graph Multiset Pooling (GMPool) without SelfAtt, where we follow the node clustering approaches as described in the subsection 3.4 by generating a single soft assignment matrix with one head $h = 1$ in the multi-head attention function. For experiments of both synthetic and molecule reconstructions, the learning rate is set to $5 \times 10^{-3}$, and hidden size is set to 32. We then optimize the network with Adam optimizer (Kingma & Ba, 2014).

**Implementation Details on Synthetic Graph**    We set the pooling ratio of all models as 25%. For the loss function, we use the Mean Squared Error (MSE) to train models. We use the early stopping criterion, where we stop the training if there is no further improvement on the training loss during 1,000 epochs. Further, the maximum number of epochs is set to 10,000. Note that, there is no other available graphs for validation of the synthetic graph, such that we train and test the models only with the given graph in the Figure 10. The baseline results are adopted from Bianchi et al. (2019).

**Implementation Details on Molecule Graph**    We set the pooling ratio of all models as 5%, 10%, 15%, and 25%, and plot all results in the Figure 6 of the main paper. Note that, in the case of molecule graph reconstruction, a softmax layer is appended at the last layer of the model architecture to classify the original atom types of all nodes. For the loss function, we use the cross entropy loss to train models. We use the early stopping criterion, where we stop the training if there is no further improvement on the validation loss during 50 epochs. Further, the maximum number of epochs is set to 500, and batch size is set to 128. Note that, in the case of molecule graph reconstruction on the ZINC dataset, we strictly separate the training, validation and test sets, as suggested by Dwivedi et al. (2020). We perform all experiments 5 times with 5 different random seeds, and then report the averaged result with the standard deviation. Note that, in addition to baselines mentioned in the common implementation details paragraph, we compare two more baselines: GCN with a random assignment matrix for pooling, which is adopted from Mesquita et al. (2020), and StructPool (Yuan & Ji, 2020), for the real-world molecule graph reconstruction.

**Evaluation Metrics for Molecule Reconstruction**    For quantitative evaluations, we use the three metrics as follows: *1) validity* indicates the number of reconstructed molecules that are chemically valid, *2) exact match* indicates the number of reconstructed molecules that are exactly same as the original molecules, and *3) accuracy* indicates the classification accuracy of atom types of all nodes.

### C.4    GRAPH GENERATION

**Common Implementation Details**    In the graph generation experiments, we replace the graph embedding function $f(G)$ from existing graph generation models to the proposed Graph Multiset Transformer (GMT), to evaluate the applicability of our model on generation tasks, as described

Table 4: Graph classification results on validation sets with standard deviations. All results are averaged over 10 different runs. Best performance and its comparable results ($p > 0.05$) from the t-test are marked in blod. Hyphen (-) denotes out-of-resources that take more than 10 days (See Figure 4 for the time efficiency analysis).

| | Biochemical Domain | | | | | | | Social Domain | | |
|---|---|---|---|---|---|---|---|---|---|---|
| | D&D | PROTEINS | MUTAG | HIV | Tox21 | ToxCast | BBBP | IMDB-B | IMDB-M | COLLAB |
| # graphs | 1,178 | 1,113 | 188 | 41,127 | 7,831 | 8,576 | 2,039 | 1,000 | 1,500 | 5,000 |
| # classes | 2 | 2 | 2 | 2 | 12 | 617 | 2 | 2 | 3 | 3 |
| Avg # nodes | 284.32 | 39.06 | 17.93 | 25.51 | 18.57 | 18.78 | 24.06 | 19.77 | 13.00 | 74.49 |
| GCN | $76.17 \pm 0.65$ | $77.13 \pm 0.44$ | $76.56 \pm 1.75$ | $81.27 \pm 0.92$ | $78.80 \pm 0.40$ | $65.66 \pm 0.40$ | $93.35 \pm 1.08$ | $77.93 \pm 0.28$ | $54.29 \pm 0.23$ | $83.08 \pm 0.13$ |
| GIN | $76.85 \pm 0.61$ | $78.43 \pm 0.45$ | $\mathbf{94.44 \pm 0.52}$ | $82.10 \pm 1.01$ | $78.20 \pm 0.45$ | $66.29 \pm 0.42$ | $94.64 \pm 0.36$ | $\mathbf{78.38 \pm 0.26}$ | $54.04 \pm 0.29$ | $82.19 \pm 0.25$ |
| Set2Set | $76.32 \pm 0.40$ | $77.64 \pm 0.41$ | $79.72 \pm 2.40$ | $80.07 \pm 0.93$ | $79.13 \pm 0.75$ | $66.39 \pm 0.49$ | $91.89 \pm 1.48$ | $78.13 \pm 0.30$ | $54.39 \pm 0.19$ | $82.34 \pm 0.23$ |
| SortPool | $80.68 \pm 0.59$ | $77.92 \pm 0.42$ | $81.33 \pm 3.00$ | $81.17 \pm 2.30$ | $75.97 \pm 0.76$ | $64.26 \pm 1.17$ | $94.21 \pm 1.04$ | $77.46 \pm 0.60$ | $52.95 \pm 0.62$ | $80.58 \pm 0.25$ |
| DiffPool | $81.33 \pm 0.33$ | $79.09 \pm 0.36$ | $87.94 \pm 1.93$ | $\mathbf{83.16 \pm 0.44}$ | $80.02 \pm 0.38$ | $\mathbf{69.73 \pm 0.79}$ | $\mathbf{96.32 \pm 0.36}$ | $77.86 \pm 0.39$ | $54.77 \pm 0.19$ | $81.69 \pm 0.31$ |
| SAGPool(G) | $76.73 \pm 0.80$ | $77.01 \pm 0.58$ | $88.11 \pm 1.21$ | $80.55 \pm 1.89$ | $77.03 \pm 0.76$ | $65.51 \pm 0.91$ | $\mathbf{95.59 \pm 1.22}$ | $\mathbf{78.09 \pm 0.58}$ | $53.73 \pm 0.42$ | $81.91 \pm 0.45$ |
| SAGPool(H) | $79.56 \pm 0.67$ | $77.24 \pm 0.56$ | $86.06 \pm 2.07$ | $79.21 \pm 1.50$ | $75.36 \pm 2.63$ | $64.05 \pm 0.83$ | $93.05 \pm 3.00$ | $77.11 \pm 0.46$ | $53.49 \pm 0.65$ | $80.55 \pm 0.56$ |
| TopKPool | $78.54 \pm 0.73$ | $75.47 \pm 0.90$ | $75.06 \pm 2.12$ | $79.24 \pm 1.84$ | $75.06 \pm 2.30$ | $64.56 \pm 0.56$ | $93.31 \pm 2.32$ | $76.12 \pm 0.79$ | $52.75 \pm 0.58$ | $79.94 \pm 0.86$ |
| MinCutPool | $81.96 \pm 0.39$ | $79.23 \pm 0.66$ | $87.22 \pm 1.72$ | $\mathbf{83.12 \pm 1.27}$ | $\mathbf{81.10 \pm 0.42}$ | $\mathbf{69.09 \pm 1.12}$ | $\mathbf{95.99 \pm 0.47}$ | $77.76 \pm 0.36$ | $\mathbf{54.94 \pm 0.19}$ | $\mathbf{83.37 \pm 0.18}$ |
| StructPool | $\mathbf{82.56 \pm 0.37}$ | $\mathbf{80.00 \pm 0.27}$ | $91.5 \pm 0.95$ | $81.09 \pm 1.26$ | $79.61 \pm 0.70$ | $66.49 \pm 1.59$ | $95.18 \pm 0.59$ | $77.14 \pm 0.31$ | $54.13 \pm 0.39$ | $79.90 \pm 0.18$ |
| ASAP | $81.58 \pm 0.38$ | $78.71 \pm 0.45$ | $91.33 \pm 0.65$ | $79.80 \pm 1.88$ | $77.33 \pm 1.34$ | $63.82 \pm 0.75$ | $92.96 \pm 1.09$ | $77.89 \pm 0.51$ | $\mathbf{55.17 \pm 0.33}$ | $82.11 \pm 0.33$ |
| EdgePool | $80.32 \pm 0.44$ | $79.61 \pm 0.25$ | $87.28 \pm 1.18$ | $81.84 \pm 1.32$ | $78.92 \pm 0.29$ | $66.21 \pm 0.64$ | $94.98 \pm 0.62$ | $77.50 \pm 0.25$ | $54.69 \pm 0.40$ | - |
| HaarPool | - | - | $68.22 \pm 0.86$ | - | - | - | $89.98 \pm 0.58$ | $76.72 \pm 0.60$ | $53.03 \pm 0.14$ | - |
| GMT (Ours) | $82.19 \pm 0.40$ | $\mathbf{80.01 \pm 0.21}$ | $91.00 \pm 0.82$ | $\mathbf{83.54 \pm 0.78}$ | $80.91 \pm 0.41$ | $\mathbf{69.77 \pm 0.67}$ | $95.14 \pm 0.48$ | $\mathbf{78.43 \pm 0.22}$ | $\mathbf{55.14 \pm 0.25}$ | $\mathbf{83.37 \pm 0.11}$ |

in the subsection 4.3 of the main paper. As baselines, we first use the original models with their implementations. Specifically, we use the MolGAN[2] (Cao & Kipf, 2018) for molecule generation, and Graph Logic Network (GLN)[3] (Dai et al., 2019) for retrosynthesis. For both experiments, we directly follow the experimental details of original papers (Cao & Kipf, 2018; Dai et al., 2019) for a fair comparison. Furthermore, to compare our models with another strong pooling method, we use the MinCutPool (Bianchi et al., 2019) as an additional baseline for generation tasks, since it shows the best performance among baselines in the previous two classification and reconstruction tasks.

For MinCutPool, since it cannot directly compress the all $n$ nodes into the 1 cluster to represent the entire graph, we use the following trick to replace the simple pooling operation (e.g. sum or mean) with it. We first condense the graph into the k clusters ($k = 4$) using one MinCutPool layer, and then average the condensed nodes to get a single representation of the given graph. However, our proposed Graph Multiset Transformer (GMT) can directly compress the all $n$ nodes into the 1 node with one learnable seed vector, by using the single $\text{GMPool}_1$ block. In other words, we use the one $\text{GMPool}_1$ to represent the entire graph by replacing their simple pooling (e.g. sum or mean), in which we use the following softmax activation function for computing attention weights:

$$w(\boldsymbol{Q}\boldsymbol{K}^T)_{i,j} = \frac{\exp(\boldsymbol{Q}_i\boldsymbol{K}_j^T)}{\sum_{k=1}^{n}\exp(\boldsymbol{Q}_i\boldsymbol{K}_k^T)}. \tag{14}$$

**Implementation Details on Molecule Generation** For the molecule generation experiment with the MolGAN, we replace the average pooling in the discriminator with $\text{GMPool}_1$. We use the QM9 dataset (Ramakrishnan et al., 2014) following the original MolGAN paper (Cao & Kipf, 2018). To evaluate the models, we report the validity of 13,319 generated molecules at the early stage of the MolGAN training, over 4 different runs. As depicted in Figure 8 of the main paper, each solid curve indicates the average validity of each model with 4 different runs, and the shaded area indicates the half of the standard deviation for 4 different runs.

**Implementation Details on Retrosynthesis** For the retrosynthesis experiment with the Graph Logic Network (GLN), we replace the average pooling in the template and subgraph encoding functions with $\text{GMPool}_1$. We use the USPTO-50k dataset following the original paper (Dai et al., 2019). For an evaluation metric, we use the Top-$k$ accuracy for both reaction class is not given and given cases, following the original paper (Dai et al., 2019). We reproduce all results in Table 3 with published codes from the original paper.

## D  ADDITIONAL EXPERIMENTAL RESULTS

**Validation Results on Graph Classification** We additionally provide the graph classification results on validation sets. As shown in Table 4, the proposed GMT outperforms most baselines, or

[2]https://github.com/yongqyu/MolGAN-pytorch
[3]https://github.com/Hanjun-Dai/GLN

Table 6: Quantitative results of the graph reconstruction task on reconstructing the node features and the adjacency matrix for synthetic graphs, with two different minimization objectives and error calculation metrics: $X - X^{rec}$ and $A - A^{rec}$. * indicates the model without using adjacency normalization.

| Data: | Grid Graph | | | | Ring Graph | | | |
|---|---|---|---|---|---|---|---|---|
| Objective: | $\min\|X - X^{rec}\|$ | | $\min\|A - A^{rec}\|$ | | $\min\|X - X^{rec}\|$ | | $\min\|A - A^{rec}\|$ | |
| Error Calculation: | $\|X - X^{rec}\|$ | $\|A - A^{rec}\|$ | $\|X - X^{rec}\|$ | $\|A - A^{rec}\|$ | $\|X - X^{rec}\|$ | $\|A - A^{rec}\|$ | $\|X - X^{rec}\|$ | $\|A - A^{rec}\|$ |
| DiffPool | 0.0833 | 12110194 | 0.3908 | 0.0856 | 0.0032 | 617.7706 | 0.6208 | 0.0948 |
| MinCutPool | 0.0001 | 0.0092 | 1.2883 | 0.0051 | 0.0005 | 0.0424 | 0.5026 | 0.0128 |
| MinCutPool* | 0.0002 | 201.7619 | 2.0261 | 0.0616 | 0.0003 | 68.23 | 0.5211 | 0.0725 |
| GMT (Ours) | 0.0001 | 0.0102 | 0.2353 | 0.0084 | 0.0000 | 0.0331 | 0.5475 | 0.0324 |

achieves comparable performances to the best baseline results even in the validation sets. While validation results can not directly measure the generalization performance of the model for unseen data, these results further confirm that our method is powerful enough, compared to baselines. Regarding the results of test sets on the graph classification task, please see Table 1 in the main paper.

**Leaderboard Results on Graph Classification**   For a fair comparison, we experiment with all baselines and our models in the same setting, as described in the implementation details of Appendix C.2. Specifically, we average the results over 10 different runs with the same hidden dimension (128, while leaderboard uses 300), and the same number of message-passing layers (3, while leaderboard uses 5) with 10 different seeds for all models. Therefore,

Table 5: Graph classification results for OGB test datasets with standard deviations.

| | Model | HIV | Tox21 |
|---|---|---|---|
| Leaderboard | GCN | $76.06 \pm 0.97$ | $75.29 \pm 0.69$ |
| | GIN | $75.58 \pm 1.40$ | $74.91 \pm 0.51$ |
| Reproduced | GCN | $76.81 \pm 1.01$ | $75.04 \pm 0.80$ |
| | GIN | $75.95 \pm 1.35$ | $73.27 \pm 0.84$ |
| Ours | GMT | $\mathbf{77.56} \pm 1.25$ | $\mathbf{77.30} \pm 0.59$ |

the reproduced results can be slightly different from the leaderboard results, as shown in Table 5, since the leaderboard uses different hyper-parameters with different random seeds (See Hu et al. (2020) for more details). However, our reproduction results are almost the same as the leaderboard results, and sometimes outperform the leaderboard results (See the GCN results for the HIV dataset in Table 5). Therefore, while we conduct all experiments under the same setting for a fair comparison, where specific hyperparameter choices are slightly different from the leaderboard setting, these results indicate that there is no significant difference between reproduced and leaderboard results.

**Quantitative Results on Graph Reconstruction for Synthetic Graphs**   While we conduct experiments on reconstructing node features on the given graph, to quantify the retained information on the condensed nodes after pooling (See Section 4.2 for experiments on the graph reconstruction task), we further reconstruct the adjacency matrix to see if the pooling layer can also condense the adjacency structure without loss of information. The learning objective to minimize the discrepancy between the original adjacency matrix $A$ and the reconstructed adjacency matrix $A^{rec}$ with a cluster assignment matrix $C \in \mathbb{R}^{n \times k}$ is defined as follows: $\min\|A - A^{rec}\|$, where $A^{rec} = C A^{pool} C^T$.

Then we design the following two experiments. First, pooling layers are trained to minimize the objective in Section 4.2: $\min\|X - X^{rec}\|$. After that, we measure the discrepancy between the original and the reconstructed node features: $\|X - X^{rec}\|$, and also measure the discrepancy between the original and the reconstructed adjacency matrix: $\|A - A^{rec}\|$. Second, pooling layers are trained to minimize the objective described in the previous paragraph: $\min\|A - A^{rec}\|$, and then we measure the aforementioned two discrepancies in the same way.

We experiment with synthetic grid and ring graphs, illustrated in Figure 10. Table 6 shows that the error is large when the objective and the error metric are different, which indicates that there is a high discrepancy between the required information for condensing node and the required information for condensing adjacency matrix. In other words, the compression for node and the compression for adjacency matrix might be differently performed to reconstruct the whole graph information.

Also, Table 6 shows that there are some cases where there is no significant difference in the calculated adjacency error ($\|A - A^{rec}\|$), when minimizing nodes discrepancies and minimizing adjacency discrepancies (See 0.0331 and 0.0324 for the proposed GMT on the Ring Graph). Furthermore, calculated errors for the adjacency matrix when minimizing adjacency discrepancies are generally larger than the calculated errors for node features when minimizing nodes discrepancies. These results indicate that the adjacency matrix is difficult to reconstruct after pooling. This might be because the reconstructed adjacency matrix should be further transformed from continuous values

to discrete values (0 or 1 for the undirected simple graph), while the reconstructed node features can be directly represented as continuous values. We leave further reconstructing adjacency matrices and visualizing them as a future work.

**Additional Examples for Molecule Reconstruction** We visualize the additional examples for molecule reconstruction on the ZINC dataset in Figure 11. Molecules on the left side indicate the original molecule, where the transparent color denotes the assigned cluster for each node, which is obtained by the cluster assignment matrix $C$ with node (atom) representations in a graph (molecule) (See Proposition 5 for more detail on generating the cluster assignment matrix). Also, molecules on the right side indicate the reconstructed molecules with failure cases denoted as a red dotted circle.

As visualized in Figure 11, we can see that the same atom or the similarly connected atoms obtain the same cluster (color). For example, the atom type O mostly obtains the yellow cluster, and the atom type F obtains the green cluster. Furthermore, ring-shaped substructures that do not contain O or N mostly receive the blue cluster, whereas ring-shaped substructures that contain O and N receive the green and yellow clusters respectively.

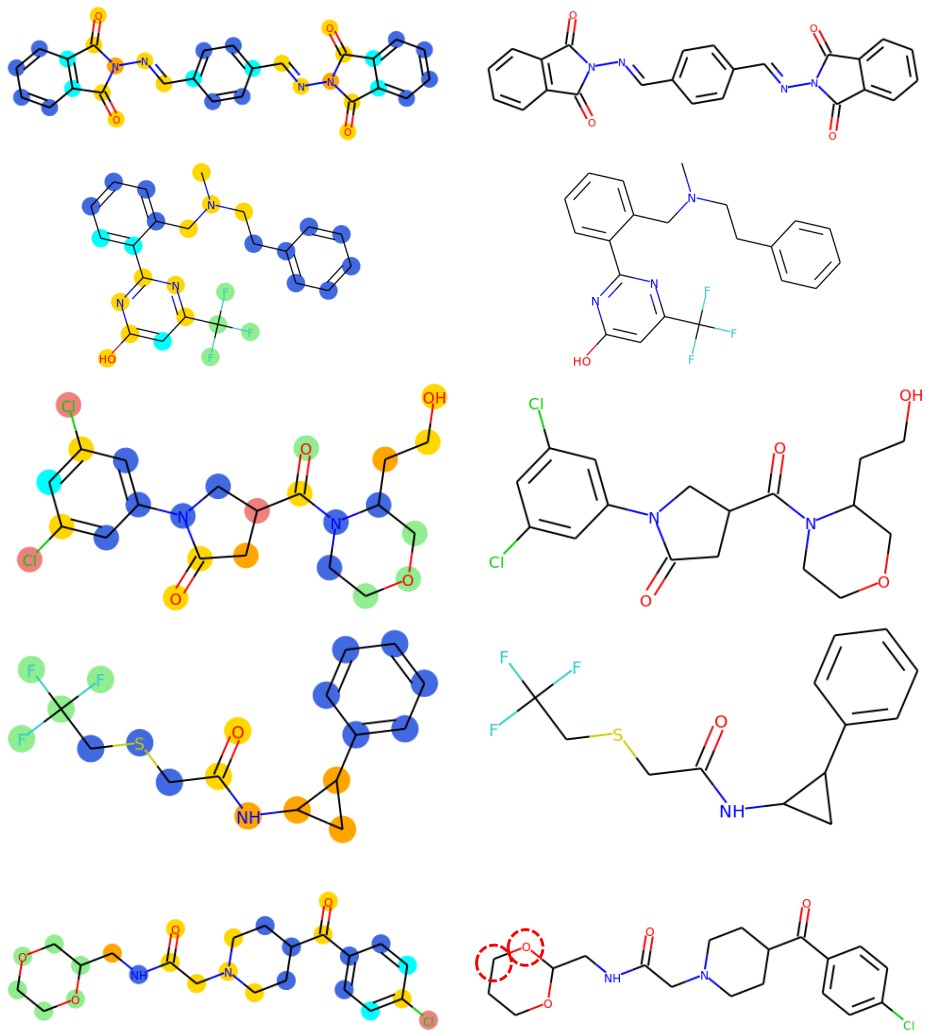

Figure 11: **Molecule Reconstruction Examples (Left):** Original molecules with the assigned cluster on each node represented as color, where cluster is generated from *Graph Multiset Pooling* (GM-Pool). **(Right):** Reconstructed molecules. Red dotted circle indicates the incorrect atom prediction.

