# OpenReview forum: "Accurate Learning of Graph Representations with Graph Multiset Pooling"
_ICLR.cc/2021/Conference — ICLR 2021 Poster_

### Official Review · AnonReviewer1 · 2020-10-26
**A method for multi-head attention pooling on graphs**

**Rating:** 7
**Confidence:** 4

**Review:**

The work extends the set transformer to obtain a method for multi-head attention pooling on multisets with connectivity (graphs). The authors show that the approach is as expressive as the WL isomorphism test and has better space complexity than existing node clustering networks. The method achieves state-of-the-art results on graph classification and strong results on graph reconstruction and generation.

Strengths:
- The paper is very well written and polished.
- The figures complement the text well.
- The work is technically and mathematically sound.
- The method shows good results and is scalable, making it a valuable addition to the set of existing GNN operators.
- The authors make an effort to substantiate their statements about expressivity and scalability with proofs.
- The experiments are well chosen and show where improvements come from.

Weaknesses:
- The proven expressiveness is not a very strong statement, since most pooling approaches adhere to this property. It is nice to have the theoretical analysis though.
- The method itself is an incremental variation of set transformers (Lee et al.), although adapted for a different type of input data.
- Using the identity matrix as adjacency (as described in Appendix B, to work around the scalability issue of node clustering methods) seems to make the approach identical to the set transformer in all layers except the first, which dampens the contribution.

There is some potential for improvement in clarity of presentation:
- In abstract: "may yield representations that do not pass the graph isomorphism test", in introduction, page 2: "accurate representations of given graphs that can pass the WL-test". Those sentences are confusing as it is not about representations passing the WL test, is it? It is about two graphs which are distinguished by WL get different representations (as correctly stated elsewhere in the work).
- Regarding page 4, paragraph "Graph Multi-head Attention" and the following ones:
	- On the one hand, they are extremely close to Vaswani et al. and Lee et al. (sometimes even nearly the same sentences).
	- On the other hand, some things are left out, which are crucial for understanding, such as definitions of symbols for dimensionalities (n_q, d_k, d_v, d_model) and the origin of some matrices, see next point.
	- Where do the seeds S come from (for the non self attention operator)? It probably is a parameter matrix that is directly optimized but it is not completely clear ("learnable" is ambiguous, can also be the output of a network)
	- Suggestion: maybe the description of the pooling method becomes more clear when described in a top-down manner: Eq 7 -> Eq 6 -> Eq 5 -> Att


Experiments:
- Variances of graph classification results over the cross validation would be greatly appreciated (since there seems to be a space issue, they can go into the appendix)
- The reconstruction architecture does not reconstruct adjacency. It might be interesting to see how well the method can do that (for the synthetic graphs for example)

Related work:
- The work [1] should be mentioned and discussed in related work and compared against in experiments. It is also a pooling method with attention but seems to follow a different approach.

Typos:
- Figure 2 caption: "to compress the all nodes" -all
- Proof of Theorem 4: "but it is highly efficient than node clustering methods"
- Proof of Proposition 5: "then the first them inherently generates"

All in all, I think this paper has a valuable contribution, even if the method is incremental. Therefore, I tend to vote for accepting the paper but encourage the authors to improve on the mentioned issues in experiments, related work and presentation.

[1] Ranjan et al.: ASAP: Adaptive Structure Aware Pooling for Learning Hierarchical Graph Representations, AAAI 2020

---

> ### Author Response · Authors · 2020-11-14
> **Initial Response to R1**
>
> We sincerely appreciate your constructive and helpful comments. We initially address all your comments below:
>
> **Question 1:** In abstract: "may yield representations that do not pass the graph isomorphism test", in introduction, page 2: "accurate representations of given graphs that can pass the WL-test". Those sentences are confusing as it is not about representations passing the WL test. It is about two graphs which are distinguished by WL get different representations (as correctly stated elsewhere in the work).
>
> **Answer:** We apologize for the confusion, and thank you for pointing out the inaccurate wording. The two sentences in the abstract and introduction mean that the representation power for the proposed method is as powerful as the WL test, such that the proposed pooling method represents the two distinct graphs differently, while the baseline may yield the same representation for two different graphs distinguished by the WL test. We have corrected them in the revision.
>
> ---
>
> **Question: 2** Some things are left out in the Graph Multi-head Attention paragraph, such as definitions of symbols for dimensionalities (n_q, d_k, d_v, d_model). Also, where do the seeds S come from?
>
> **Answer:** We apologize for the confusion, and thank you for pointing them out. We carefully revised the description of the Graph Multi-head Attention for improved clarity. The seed matrix S is a parameter matrix that is directly optimized, and not an output of a network.
>
> ---
>
> **Question 3:** Variances of graph classification results over the cross validation would be greatly appreciated.
>
> **Answer:**  We have reported the standard deviation on graph classification results with a p-value for the significance test in Table 1 (graph classification), and also visualized the variance in Figure 6 (graph reconstruction) of the revision, as suggested.
>
> ---
>
> **Question 4:** The reconstruction architecture does not reconstruct adjacency. It might be interesting to see how well the method can do that.
>
> **Answer:** Thanks for your suggestion, and we have **included and discussed the results on reconstructing the adjacency matrix**from the compressed representation in Appendix D, Adjacency Reconstruction for Graph Reconstruction paragraph of the revision. Please note that reconstruction for the adjacency matrix is more difficult than reconstruction for nodes and thus are not covered in any of the existing works to our knowledge , since, while the reconstructed node feature can be represented as continuous values, the reconstructed adjacency matrix should be further transformed from continuous values to discrete values (0 or 1 for the undirected simple graph). We thus leave the reconstruction of the adjacency matrices as future work.
>
> ---
>
> **Question 5:** The work [1] should be mentioned and discussed in related work and compared against in experiments. It is also a pooling method with attention but seems to follow a different approach.
>
> **Answer:** ASAP [1] is clearly different from the proposed GMPool. ASAP first computes the local cluster assignment using the message-passing operation, and then drops the unimportant local clusters with their scores. However, the proposed GMPool globally compresses the $n$ nodes into $k$ nodes with their relevance score for the seed matrix S, and does not drop any nodes, and thus does not suffer from any information loss. We have compared against ASAP, and you can see its result in the response to the common comments and Table 1.
>
> ---
>
> **Question 6:** Typos.
>
> **Answer:** Thanks for pointing them out. We have corrected them in the revision.
>
> ---
>
> [1] Ranjan et al.: ASAP: Adaptive Structure Aware Pooling for Learning Hierarchical Graph Representations, AAAI 2020

---

> ### Author Response · Authors · 2020-11-23
> **The end of the discussion phase approaching, and we address all your comments.**
>
> Dear AnonReviewer 1,
>
> We sincerely appreciate your positive comments that our paper is very well written, technically as well as mathematically sound with theoretical justification, and the proposed method shows good results with scalable complexity, making it a valuable addition to the set of existing GNN operators. During the rebuttal period, we have made every effort to faithfully address all your and other reviewers’ comments in the response as well as in the revision. Here, we briefly summarize the major updates as follows:
>
> * We have corrected the inaccurate wording regarding the WL-test, and the definitions of symbols in the graph multi-head attention paragraph. Thank you for pointing them out.
> * We have compared the suggested ASAP baseline in Table 1, and included the standard deviations in Table 1 and Figure 6 of the revision.
> * We have discussed the reconstruction of the adjacency matrix in Table 6 of the revision.
>
> We sincerely appreciate your insightful and constructive comments.
>
> Thanks, Authors

---

> ### Author Response · Authors · 2020-11-25
> **The interactive discussion phase ending in 7 hours**
>
> Dear reviewer,
>
> We thank you for your helpful and constructive comments. We have done our best to respond to your comments as well as other reviewers’ comments in the discussion phase, and have faithfully reflected comments in the revision, which made our paper stronger. Please let us know if there is anything else we should address, since we cannot have interactions with you past the deadline, which is less than 7 hours away. We thank you so much again for your time and effort to review our paper.
>
> Best regards,
> Authors

---

### Official Review · AnonReviewer2 · 2020-10-27
**Official Blind Review #2**

**Rating:** 6
**Confidence:** 5

**Review:**

This work studies the graph pooling operation for graph neural networks. It proposes the Graph Multiset Pooling which treats the graph pooling as a multiset encoding problem and can capture the graph structural information. It first employs multi-head attention to learn node features, where the query Q is a learnable matrix contains k vectors. Then a GMPool operation is performed and finally, the self-attention is used for learning inter-node relationships. Experimental results show the effectiveness of the proposed method.

Strengths:
+ This work studies an important problem, graph pooling. Graph pooling can learn high-level graph representations but is still less explored.
+ The proposed method is interesting. By using a learnable query matrix, the method can reduce the n-node input to k-node output. The self-attention used after GMPool can learn the relationships between high-level embeddings.
+ The experimental results are promising. The proposed method outperforms other compared methods.

Weaknesses:
- Even though the method is called Multiset Pooling, its method is not related to Multiset. The proposed method is mainly based on attention and self-attention mechanism. Then claiming the proposed method as Multiset Pooling is not convincing.
- The experimental settings are not fair enough. The pooling operation is defined as reducing n-node input to k-node output. For all other methods, the pooling layer is connected with the global sum/average. However, in the proposed GMT, the GMPool is connected with a self-attention layer. It is not clear whether the proposed GMPool or the self-attention layer leads to the performance gain? A careful ablation study is needed.
- I think the proposed method can be regarded as using SAGPool in the clustering-based pooling. The main difference is the multi-head attention and the learnable matrix S. Please comment if I missed something.
- The use of graph structures is not very convincing. The GNN(H, A) is the simple message passing of GNNs. Then the graph structural information A is already incorporated in H since H is obtained by GNNs.
- Several baselines are missing, such as Structpool and Edge Pooling. They should be discussed and compared.

Questions:
1. If the query Q is obtained from the learnable matrix S, which is k*d dimensions. Then the output of  Att(Q; K; V ) should also have k*d’ dimension, which means the output is already reduced from n vectors to k vectors. Why do we still need the GMPool operation? The GMPool does not “compress the n nodes into the k typical nodes”.


=====Update after rebuttal=====

I have read the authors' rebuttal. Most of my concerns are addressed properly, and hence I am willing to increase my score from 4 to 6.

---

> ### Author Response · Authors · 2020-11-14
> **Initial Response to R2 (2/2)**
>
> **Question 3:** I think the proposed method can be regarded as using SAGPool in the clustering-based pooling. The main difference is the multi-head attention and the learnable matrix $S$.
>
> **Answer:** This is a critical misunderstanding since our method and SAGPool have little in common, even when not considering the fact that our method captures both a multiset and a global graph structure among the nodes. SAGPool refers to the **message-passing operation as the self-attention**in equation 1, and they are not conventional self-attention mechanisms. The main idea of SAGPool is to drop unimportant nodes for pooling, whereas the proposed GMT uses the attention to weight the node representations with their relevance to the query. Further note that GMT encodes a multiset and considers the graph structure when pooling (Figure 1(Right)).
>
> ---
>
> **Question 4:** The use of graph structures in GMPool is not very convincing since the graph structural information $A$ is already incorporated in $H$.
>
> **Answer:** While the graph structural information $A$ might be preserved in $H$, the naive attention function generates the keys and values that are linearly transformed from $H$, which is suboptimal in capturing the graph structure. To tackle these limitations, we use two separate GNNs with graph structural information $A$ to generate the key and value matrices, which can more explicitly preserve the structural information on $A$ than the simple linear transformation. We empirically show that the proposed Graph Multi-head Attention achieves significantly better results than naive multi-head attention (See Table 2 and Figure 6). We have clarified this point in Section 3.2, Graph Multi-head Attention paragraph in the revision.
>
> ---
>
> **Question 5:** More baselines, such as Structpool and Edge Pooling, should be discussed and compared.
>
> **Answer:** Thank you for the suggestion. Please note that we compare against MinCutPool (ICML 2020), which is **more recent work than StructPool**and were not aware of Edge Pooling at the time of submission since it is an Arxiv paper. We have discussed Structpool and Edge Pooling in the related work section of the revision, and further provide initial results that we obtained with them thus far (Note that Edge Pooling is very slow) in the common comment, and have also included them in Table 1, Figure 3, and Figure 4 of the revision. We will include the full experimental comparison against the two baselines in the final revision.
>
> ---
>
> **Question 6:** If the query $Q$ is obtained from the learnable matrix $S$, which is $k \times d$ dimensions. Then the output of Att($Q$; $K$; $V$ ) should also have $k \times d$ dimension, which means the output is already reduced from $n$ vectors to $k$ vectors. Why do we still need the GMPool operation?
>
> **Answer:** We require the GMPool operation, since nodes are reduced from $n$ nodes to $k$ nodes through GMPool, not by the conventional attention function which learns an $n$-to-$n$ mapping. The reason why we construct several attention functions such as Att (Attention), MH (Multi-head Attention in equation 5), and GMH (Graph Multi-head Attention in equation 6) in the Graph Multi-head Attention paragraph is that these operations are used for both GMPool and SelfAtt. In other words, these attention components are not only used for compressing nodes with GMPool but are also used for capturing global node relationships with SelfAtt. Thus, the Att function with $k \times d$ dimension is a specific design choice for compressing $n$ nodes into $k$ typical nodes, and GMPool makes this possible with the learnable matrix $S$.

---

> > ### Comment · AnonReviewer2 · 2020-11-24
> > **Response**
> >
> > Thanks for your detailed response and conducting more experiments as requested. I think some of my concerns are addressed. However, I still have some questions.
> >
> > Q2-1. For the graph classification task, most comparing methods are employing a global readout function to obtain graph-level embeddings. What is the global readout function? In addition, for your proposed method, it is claimed that "Finally, we get the entire graph representation by using GMPool with k = 1". Then for other comparing methods, if we employ the same readout function, will the performance be better than the proposed method?
> >
> > Q2-2: To my understanding, the pooling operation is finished after the GMPool, as shown in Figure 2. The other operations such as SelfAtt and GMpool(k=1) should not be considered as a part of the pooling operation. For pooling, we want to learns the mappings from n nodes to k nodes. That's why I believe the comparisons are not fair enough.
> >
> > Q3. The key idea of SAGPool is to use attention scores to select important nodes. The proposed method employs attention scores as the weights for clustering different nodes. Compared with SAGpool, I believe the novelty is limited.
> >
> > Q6. In your Equation 7, the GMH refers to the Graph Multi-head Attention layer, which takes the S as the query. It means the output of this GMH contains $k$ nodes. Then why do you still need the LN and rFF? What's the motivation?

---

> > > ### Author Response · Authors · 2020-11-24
> > > **Re: Response (2/2)**
> > >
> > > **Question A-3. The key idea of SAGPool is to use attention scores to select important nodes. The proposed method employs attention scores as the weights for clustering different nodes. Compared with SAGpool, I believe the novelty is limited.**
> > >
> > > **Answer.**
> > > This is a **critical misunderstanding**since SAGPool and GMT have very little in common, either **conceptually (node-drop vs. graph multiset transformer)**, or **component-wise (SAGPool’s attention is completely different from our attention mechanism)**. To clear up your misunderstanding, we rewrite the attention mechanism of SAGPool as follows:
> > > - $Z = \sigma (GNN (X, A))$
> > >
> > > (Please see equation 3, 4, 5, and 6 of [A]), where they use $Z \in \mathbb{R}^{n \times 1}$ with n nodes as the attention scores, which are used to further drop unimportant nodes. Thus, the attention they refer to is the output of the GNN squashed into the 1-dimensional value, and is completely different from the self-attention mechanisms used in our pooling function even in the most basic building block.
> > >
> > > On the other hand, we use the attention function that has the query $Q$, key $K$ and value $V$, to first calculate the relevance scores between the query $Q$ and the key $K$ matrices, and then compute the weighted sum of values $V$ with calculated relevance scores, formalized as follows (Please see $Att(Q, K, V)$ in Graph Multi-head Attention paragraph of section 3):
> > > - $Att(Q, K, V) = w(Q K^T) V$.
> > >
> > > Thus, **even the basic attention mechanisms are different**.
> > >
> > > Note that we further use:
> > > - **Multi-head attention**function $MH(Q, K, V)$ in equation 5, to extend the naive attention function $Att(Q, K, V)$, to yield $h$ different representation subspaces.
> > > - Newly proposed **Graph multi-head attention**function $GMH(Q, K, V)$ in equation 6 to explicitly consider graph structure into account.
> > >
> > > With these attention functions that require specific query, key, and value matrices, 1) we compress the $n$ nodes into the $k$ nodes with seed vector $S$ as a query matrix in GMPool (See equation 7), and we consider the **inter-node relationships**between k nodes by setting the same query, key, and value matrices in SelfAtt (See equation 8).
> > >
> > > We further report the large performance gain of our method over SAGPool (Table 1) below:
> > >
> > >
> > > | | D&D | PROTEINS | MUTAG | HIV | Tox21 | ToxCast | BBBP | IMDB-B | IMDB-M | COLLAB |
> > > | --- | --- | --- | --- | --- | --- | --- | --- | --- | --- | --- |
> > > | SAGPool (G) | 71.54 | 72.02 | 76.78 | 74.56 | 71.10 | 59.88 | 65.16 | 72.16 | 49.47 | 78.85 |
> > > | SAGPool (H) | 74.72 | 71.56 | 73.67 | 71.44 | 69.81 | 58.91 | 63.94 | 72.55 | 50.23 | 78.03 |
> > > | GMT (Ours) | **78.72** | **75.09** | **83.44** | **77.56** | **77.30** | **65.44** |**68.31**| **73.48**| **50.66**| **80.74**|
> > >
> > > It is also important to note that node drop methods, including SAGPool and TopKPool, generally underperform other pooling baselines (Further see Figure 5 and Figure 6), since they unnecessarily drop arbitrary nodes.
> > >
> > > ---
> > >
> > > [A] Lee et al., “Self-Attention Graph Pooling.” ICML 2018.
> > >
> > > [B] Vaswani et al. “Attention Is All You Need.” NIPS 2017.
> > >
> > > [C] Hu et al. “Open Graph Benchmark: Datasets for Machine Learning on Graphs.” NeurIPS 2020.
> > >
> > > [D] Xu et al. “Understanding and Improving Layer Normalization.” NeurIPS 2019.

---

> > > ### Author Response · Authors · 2020-11-24
> > > **Re: Response (1/2)**
> > >
> > > We sincerely appreciate your helpful feedback. We respond to your comments below:
> > >
> > > **Question A-2-1 (1). For the graph classification task, most comparing methods are employing a global readout function to obtain graph-level embeddings. What is the proposed global readout function?**
> > >
> > > **Answer.** As clearly illustrated in **Figure 1 and Figure 2**, the proposed global readout function learns to map the **$n$ nodes into the entire graph representation**, on which we compress the elements over the multiset with GMPool in equation 7 and, we consider the inter-relationships between nodes with SelfAtt in equation 8.
> > >
> > > ---
> > >
> > > **Question A-2-1 (2). In addition, for your proposed method, it is claimed that "Finally, we get the entire graph representation by using GMPool with k = 1". Then for other comparing methods, if we employ the same readout function, will the performance be better than the proposed method?**
> > >
> > > **Answer.** Not only ours, but other baseline global pooling methods (e.g. SortPool, StructPool, Set2Set) also use their own parametric pooling function to accurately get an entire graph representation. The performances of GCN and GIN reported on the OGB dataset (See Table 15 of [C]), also use the virtual nodes, which is one of the global pooling schemes. Using the parameterized global pooling is an important research direction, and there is no reason not to use this parametrized global pooling if it improves the performance, since it adds a marginal amount of parameters and memory overhead (See the caption of Figure 1: “Grey box indicates the readout layer, which is compatible with our method”, 13) GMT of Appendix C.1 Baselines and our model: “it can be coupled with any pooling methods by replacing their last layer”, and the memory efficiency results in Figure 3).
> > >
> > > ---
> > >
> > > **Question A-2-2. To my understanding, the pooling operation is finished after the GMPool, as shown in Figure 2. The other operations such as SelfAtt and GMpool(k=1) should not be considered as a part of the pooling operation. For pooling, we want to learn the mappings from n nodes to k nodes. That's why I believe the comparisons are not fair enough.**
> > >
> > > **Answer.**
> > > * This seems like a misunderstanding coming from confusing a global pooling operation (Graph Multiset Transformer) with a hierarchical pooling operation.
> > > - The **global pooling**operation learns to map $n$ nodes into a single graph representation.
> > > - The **hierarchical pooling**operation learns to recursively map $n$ nodes into $k$ representations.
> > > Thus, what you are referring to is a hierarchical pooling operation, but the case is different for a global pooling operation (Please see Figure 1 and Figure 9 for high-level illustrations, and Appendix C.1 for detailed model explanations).
> > >
> > > * Graph Multiset Transformer is a global pooling scheme which globally represents the entire graph with $n$ nodes into **a single global representation**, by considering graph structure (GMPool in Figure 2) and inter-node relationships (SelfAtt in Figure 2) into account. Thus, the SelfAtt and GMPool(k=1) should exist as core components of the proposed global pooling function, namely Graph Multiset Transformer.
> > >
> > > * To summarize these, we propose the global pooling function, which we refer to as Graph Multiset Transformer, and the SelfAtt and GMPool(k=1) should be included as core components of the proposed function, as the former considers the inter-relationships and the latter further compresses the few nodes into one graph representation. We want to also emphasize that GMPool alone can be incorporated into the hierarchical pooling scheme, where the function **maps $n$ nodes to $k$ nodes**, as we describe in proposition 5 and Appendix B.
> > >
> > > * Further, as shown in Figure 5, Figure 6 and Figure 7, when incorporating GMPool into a hierarchical pooling scheme, it significantly outperforms other hierarchical pooling baselines, such as TopKPool, DiffPool and MinCutPool, and these results clearly confirm that the proposed pooling component, namely GMPool, captures the structural information of a graph well, when compressing the n nodes into the k nodes.
> > >
> > > ---
> > >
> > > **Question A-6. In your Equation 7, the GMH refers to the Graph Multi-head Attention layer, which takes the S as the query. It means the output of this GMH contains k nodes. Then why do you still need the LN and rFF? What's the motivation?**
> > >
> > > **Answer.**
> > > As you mentioned, the output of GMH already contains k nodes. However, we use the Layer Normalization and row-wise Feed Forward networks, for better optimization of the attention architecture motivated by the transformer network [B] (See Table 1 of [D] for detailed discussions with ablation results of Layer Normalization on Transformer).

---

> > > ### Author Response · Authors · 2020-11-25
> > > **The interactive discussion phase will end in less than 7 hours**
> > >
> > > Dear reviewer,
> > >
> > > We have done our best to respond to your comments below. We have clarified that our GMT is completely different from SAGPool, not just for its ability to capture a multiset with repeating elements as well as the global structure among the node embeddings, but also in the basic attention mechanisms themselves. Also, we have clarified that our method is a global pooling rather than a hierarchical pooling operation, and that GMPool is one of the components of our GMT that can be applied to hierarchical pooling. Please let us know if there is anything else we should address, since we cannot have interactions with you past the deadline, which is less than 7 hours away. We thank you so much again for your helpful comments as well as your discussions while interacting with us.
> > >
> > > Best regards,
> > > Authors

---

> ### Author Response · Authors · 2020-11-14
> **Initial Response to R2 (1/2)**
>
> We sincerely appreciate your constructive and helpful comments. We initially address all your comments below:
>
> **Question 1:** Even though the method is called Multiset Pooling, its method is not related to Multiset. The proposed method is mainly based on attention and self-attention mechanisms. Then claiming the proposed method as Multiset Pooling is not convincing.
>
> **Answer:** This is a critical misunderstanding, as the proposed method is **not**a straightforward application of the Transformer architecture. We clearly show that such a method cannot capture a graph with two identical node embeddings in **Figure 1(Right)**, and propose a **multiset transformer**to capture recurring elements with identical values, and further consider the **connectivities between the nodes**to capture the global graph structure.
>
> - As clearly shown in Figure 1(Right) and stated in the Multiset Encoding paragraph in Section 3.2: Graph Multiset Pooling, the nodes in a graph may form a multiset (i.e. a set with repeating elements) since different nodes can have identical feature vectors. If we ignore the property of the multiset in a graph, then we cannot uniquely embed two different graphs into two distinct embeddings that can be distinguished by the WL test, which is highly suboptimal. Actually, one of the main argument of this paper is that pooling layers should be designed to capture a multiset.
>
> - We further **theoretically prove**that the proposed GMT is an injective function over the multiset scheme, to obtain the graph representation that is as powerful as the Weisfeiler-Lehman (WL) test for distinguishing two different graphs (Please see Theorem 1, Lemma 2, and Proposition 3 in Section 3.3).
>
> - Moreover, a major drawback of the naive multi-head attention is that it can not explicitly reflect the graph structural information since it linearly projects the obtained node embeddings to generate the key and value. To this end, we further propose a novel graph multi-head attention block to consider graph structure into account (See Graph Multi-head Attention paragraph in Section 3.2) that is clearly different from the conventional attention mechanism.
>
> ----
>
> **Question 2-1:** The experimental settings are not fair enough. For all other methods, the pooling layer is connected with the global sum/average.
>
> **Answer:** This is a critical misunderstanding, as **not all baseline pooling methods use global sum/average**over the node embeddings. As clearly illustrated in Figure 1 and Figure 9, the architectural designs for graph representation learning tasks are broadly distinguished into global and hierarchical schemes, and the global schemes replace the sum/average operations, including Set2Set, SortPool and SAGPool(G), that are described in Figure 9 and the Experimental Setup section in Appendix C.1. Following this conventional setting, the proposed GMT is defined under the global pooling scheme by taking one global pooling operation that reduces the n nodes input to a single graph representation output. Therefore, it is fairly compared with other models.
>
> ----
>
> **Question 2-2:** GMPool is connected with a self-attention layer.
>
> **Answer:** GMPool does not use a self-attention layer for graph reconstruction, which we believe is a **more direct task to validate the expressiveness**of a graph pooling method as it can quantify the amount of the retained graph information after pooling. On the other hand, for graph classification, identifying discriminative features might be more important than the accurately capturing the structures of graphs. As clearly stated in the experimental setting for graph reconstruction in Appendix C.3, GMPool is not connected with a self-attention layer, and we use one pooling layer for all compared models. Also, Figure 5, Figure 6, and Figure 7 show that GMPool **significantly outperforms**all other compared models in the reconstruction task, and these results confirm that the proposed GMPool alone captures the structural information of a graph well when pooling the graph.
>
> ----
>
> **Question 2-3:** It is not clear whether the proposed GMPool or the self-attention layer leads to the performance gain.
>
> **Answer:**  First of all, the self-attention layer is an essential part of our Graph Multiset Tranformer. Moreover, the ablation study in Table 2 shows that all components of our graph multiset transformer **does contribute**to the performance gains on the three datasets (D&D, PROTEINS, and BBBP) we consider for graph classification. Further note that we do not use the self-attention layer for the graph reconstruction tasks, on which our method obtains incomparably higher performances over the baselines. Thus our method can accurately represent the graph without the self-attention layer.

---

> ### Author Response · Authors · 2020-11-23
> **The end of the discussion phase approaching, and we address all your comments.**
>
> Dear AnonReviewer 2,
>
> We appreciate your positive comments that we study the important but less explored graph pooling problem, the proposed pooling method is interesting that can reduce the n-node input to k-node output with further considering relationships between nodes, and that the experimental results are promising. During the rebuttal period, we have made every effort to faithfully address all your comments in both the responses and the revision. Here, we briefly summarize the major updates as follows:
>
> * We have clarified your misunderstanding that the proposed method does not encode a multiset. We also further showed that the GMT works well even without a self-attention layer as shown in Figure 5, Figure 6, and Figure 7 of the reconstruction results.
> * We have revised the graph multi-head attention paragraph to clarify the advantage of the graph multi-head attention function that can explicitly consider the graph structures on generating the key and the value matrices (Please see Table 2 and Figure 6 for the ablation results).
> * We have compared against the two baselines you have suggested, namely StructPool and EdgePool (Please see Table 1, Figure 3, Figure 4, Figure 6).
>
> Could you please go over our full responses and the revision since we can have interactions with you only by this Tuesday (24th)? We sincerely appreciate your insightful and constructive comments.
>
> Thanks, Authors

---

### Official Review · AnonReviewer3 · 2020-10-29
**This work proposes a multi-head attention-based approach to capture node interactions in improving graph pooling. While it contributes a few novel ideas, several parts of the manuscript need more explanations. Additional experimental results are needed.**

**Rating:** 4
**Confidence:** 4

**Review:**

This work proposes a Graph Multiset Transformer (GMT) that uses a multi-head attention-based approach to capture potential interactions between nodes when pooling nodes to produce a graph representation. Multi-head attention mechanism is used to group nodes into clusters, each of which produces a representation. Self-attention is then used to pool representations of clusters into the representation of a graph.

Pros:
The proposed pooling is more reasonable than a simple sum or average pooling as the multi-head attention mechanism can potentially capture dependencies between nodes.

Cons: Several parts of the manuscript need more explanations. Additional experimental results (see below for details) are needed.
(1) It is not clear what multi-head attention achieve semantically (as claimed: better capture structure information).
(2) Figures are neither self-explained nor well explained in the main text.
(3) The std of cross validation in each experiment should be reported. The mean performance alone is not enough to say a method performs better or not. It will be better to provide a p-value (e.g., t-test) to show if a method is statistically significantly better.
(4) Four datasets from the Open Graph Benchmark were used. The authors should refer to the leaderboard for the performance of some baseline methods. For example, the leaderboard of HIV dataset reports that GIN has 0.7654 rather than 0.7595 listed in this manuscript.
(5) The abstract points out that without considering task relevance is a weakness of the previous pooling method. However, it is not clear how the proposed approach make improvement(s) on this.
(6) The proposed method does not necessary pass graph isomorphism as the nodes in the manuscript have attributes but the proof does not consider node attributes.
(7) In appendix, examples in Figure 10 are confusing. More explanations are needed.

---

> ### Author Response · Authors · 2020-11-14
> **Initial Response to R3 (2/2)**
>
> **Question 4:**  Four datasets from the Open Graph Benchmark were used. The authors should refer to the leaderboard for the performance of some baseline methods.
>
> **Answer:** We reproduced all baselines for a **fair comparison of all pooling models**and not the performance of message-passing layers. We ran all experiments for all baselines and our models in the **same setting**as described in appendix C.2. Specifically, we average the results over 10 different runs with the same hidden dimension (128, and leaderboard uses 300), and the same number of message-passing layers (3, and leaderboard uses 5) with 10 different seeds for all models, which is clearly described in the appendix C.2 implementation details. Therefore, the results can be slightly different from the leaderboard results (Please see the table below), since the leaderboard uses different hyperparameters with different random seeds compared to the experimented pooling models. Note that our reproduction sometimes outperforms the leaderboard results (See the GCN results on HIV). However, as suggested, we have included the leaderboard results in the revision (See Appendix D).
>
> | | HIV | Tox21 |
> | --- | --- | --- |
> | GCN | 76.81 | 75.04 |
> | GCN (Leaderboard) | 76.06 | 75.29 |
> | GIN | 75.95 | 73.27 |
> | GIN (Leaderboard) | 75.58 | 74.91 |
> | GMT (Ours) | **77.56** | **77.30** |
>
> ---
>
> **Question 5-1:** The abstract points out that without considering task relevance is a weakness of the previous pooling method.
>
> **Answer:** This is a critical misunderstanding. We clearly state in the abstract that inconsideration of task relevance is a weakness of the simple sum/average function, not the previous learnable parametric pooling methods that are optimized in an end-to-end fashion.
>
> We stated this point in the introduction section as follows:
> * As a simplest approach for graph pooling, we can average or sum all node features in the given graph. However, since such simple aggregation schemes treat all nodes equally without considering their relative importance on the given tasks, they can not generate a meaningful graph representation in a task-specific manner.
>
> Furthermore, we stated this point in the related work section as follows:
> * While averaging all node features is directly used as simplest pooling methods, simplest pooling methods result in a loss of information since they consider all node information equally without considering key features for graphs.
>
> Therefore, we did not claim that previous pooling methods, except the flat pooling method such as average/sum, could not consider the task relevance.
>
> ---
>
> **Question 5-2:** It is not clear how the proposed approach makes improvement(s) on this sum/average pooling.
>
> **Answer:** Please note that all existing parameterized pooling operations that are learnable in an end-to-end fashion, including ours, can treat nodes differently according to their task relevance, rather than considering all nodes equally as in simple sum/average pooling. This is done by the trained networks allocating more weights on the important nodes for the specific task. Figure 7 shows one specific example that different nodes have different cluster weights for the graph reconstruction task.
>
> ---
>
> **Question 6:** The proposed method does not necessarily pass graph isomorphism as the nodes in the manuscript have attributes but the proof does not consider node attributes.
>
> **Answer:** We want to clarify that the proposed method can be at most **as powerful as the WL test**for the graph isomorphism test, rather than directly passing the graph isomorphism test. To clarify this point, we have **revised the text to tone down**on the claim, as suggested. Note that the proposed Graph Multiset Pooling can uniquely map each multiset $H$ with a bounded size, such that whether a node has attributes or not does not matter for establishing the proposed pooling function that is as powerful as the WL test. Specifically, there exists a mapping function $f$ that maps nodes to prime numbers, and the summation over nodes in a multiset constitutes a unique mapping with a function $f$, as shown in our proof of Lemma 2.
>
> ---
>
> **Question 7:** In Appendix, examples in Figure 10 (Figure 11 in the revision) are confusing. More explanations are needed.
>
> **Answer:** Thanks for the suggestion. We provide more explanations about the reconstruction examples in Figure 10 (Figure 11 in the revision), including the detailed descriptions of components embedded, and the generated cluster assignments for each node or a set of nodes.

---

> ### Author Response · Authors · 2020-11-14
> **Initial Response to R3 (1/2)**
>
> We sincerely appreciate your constructive and helpful comments. We initially address all your comments below:
>
> **Question 1:** It is not clear what multi-head attention semantically better captures the structure information.
>
> **Answer:** This is a critical misunderstanding, since we do not mention that multi-head attention can better capture the structural information. We clearly state in the introduction that the set transformer cannot distinguish the two different graphs with multiple identical node embeddings nor accurately captures the graph structure among the nodes (Please see Figure 1 (Right)). Our main contribution is in addressing such limitation with the existing set transformer by proposing a **graph multiset transformer**that not only can distinguish two graphs with a different number of identical node embeddings as powerfully as the WL test, but also accurately captures the global graph structure by considering the node connectivity in graph pooling. Please see the introduction, Figure 1, and Section 3.2 for more discussions on this part.
>
> ---
>
> **Question 2:** Figures are neither self-explained nor well explained in the main text.
>
> **Answer:** We had to cut the texts in the captions for the Figures in the experiments section due to the page limit. However, we have **added them back to the revision**by reflecting your suggestion as follows:
>
> * We have revised all captions for tables and figures in the experiment section. Specifically, we have revised the captions for Table 1, Table 2, Figure 3, Figure 5, Figure 6, Figure 7, Figure 8, and Table 3, in the revision.
>
> * Note that we have refered to Figure 9 for the illustration of the model architectures in the "Results" paragraph in Section 4.1: Graph Classification, of the revision.
>
> * We have included more descriptions of the types of ablation study we conducted in the "Ablation Study" paragraph in Section 4.1: Graph Classification, of the revision.
>
> * We have included more detailed descriptions of the experimental setup in the revision, such as the compression ratio and the evaluation metrics, in the "Results on Molecule Graph" paragraph in Section 4.2: Graph Reconstruction.
>
> * We have included more descriptions of the components in Figure 7 in the main text of the revision, in the Qualitative Analysis paragraph in Section 4.2: Graph Reconstruction.
>
> Please let us know if you find anything else we should further clarify regarding the figure captions.
>
> ---
>
> **Question 3:** The std of cross-validation in each experiment should be reported. Also, it will be better to provide a p-value (e.g., t-test) to show if a method is statistically significantly better.
>
> **Answer:** We suppressed them due to space limitation, but we have added the **standard deviations**and the **p-value for the T-test**in Table 1 of the revision. We have also included the standard deviations for the graph reconstruction experiments in Figure 6. Also note that on **graph reconstruction and generation**tasks which we believe can better evaluate the expressiveness of the pooling method, our GMT is incomparably superior over the baselines.

---

> ### Author Response · Authors · 2020-11-23
> **Response to R3 about the further question, regarding the Weisfeiler-Lehman test.**
>
> **Question A-1. In the Weisfeiler-Lehman test, the nodes do not have attributes or features. The nodes in the datasets used in this paper have attributes, which make the problem more complex.**
>
> **Answer A-1.** This is a critical misunderstanding, as WL-test can be applied on the graphs with or without the node attributes (Please see the Figure 2, a-d of [A], which illustrates the Weisfeiler-Lehman Test). We recap the WL test as follows: WL test aggregates the attributes of nodes with their neighborhood, and hashes the aggregated attributed multiset into a unique attribute. Thus, WL-test actually requires node attributes to distinguish two different graphs, and the initial node attribute can be obtained by either of the three methods (a, b, c):
>
> * a) If the nodes do not have attributes, we can simply consider that all nodes have the identical attribute.
> * b) If the nodes do not have attributes, we can alternatively assign the degree of the node as an initial node attribute (See original WL test paper [C], and algorithm 1 of [A]).
> * c) If the nodes have attributes, we can directly run the WL test using the node attributes as is without having to assign artificial attributes to the nodes (See Figure 2, a-d of [A]).
>
> Thus, nodes having attributes does not make the problem more complex, and it is slightly more efficient than the non-attributes graph setting, since we can skip the parts where we assign arbitrary attributes to the nodes, as done in a) and b).
>
> ---
>
> **Question A-2. What is the value of showing GMT "is at most as powerful as the Weisfeiler-Lehman graph isomorphism test"?**
>
> **Answer A-2.** We provide the **theoretical guarantee**that the proposed GMT is as powerful as the WL test, such that the GMT can map two different graphs, that is distinguished by the WL test, into two **distinct representations**(Please see Figure 1 (right), Theorem 1, Lemma 2 and Proposition 3). Note that, this theoretical property supports all empirical results: GMT outperforms baseline graph pooling baselines in a number of graph representation learning tasks, such as graph classification, reconstruction and generation, with very large gains in the graph reconstruction task that more directly measures the amount of graph information captured by the learned representations.
>
> ---
>
> [A] Shervashidze et al. “Weisfeiler-Lehman Graph Kernels.” JMLR 2011.
>
> [B] Xu et al. “How Powerful are Graph Neural Networks?.” ICLR 2019.
>
> [C] Weisfeiler & Leman “THE REDUCTION OF A GRAPH TO CANONICAL FORM AND THE ALGEBRA WHICH APPEARS THEREIN” 1968.

---

> > ### Author Response · Authors · 2020-11-23
> > **Response to R3 about the further question, regarding the Weisfeiler-Lehman test. (2)**
> >
> > **Question B-1. Your approach only considers discrete node labels (as in the W-L test). Does not consider the scenario that each node has multiple continuous attributes?**
> >
> > **Answer B-1.** Our approach only considers discrete node labels, and we clearly assume that the input feature space $\mathcal{H}$ for the proposed pooling function is a countable set (See Lemma 2). However, which labels nodes have are less important, since we work with obtained node embeddings from the specific message-passing function [B], rather than the raw node labels. Please note that we propose a **graph pooling**function to generate a graph-level representation, **not a message passing function**for the node embeddings (See Theorem 1, and its next sentence).
> >
> > ---
> >
> > **Question B-2. Your approach aims to distinguish "two different graphs" that are distinguishable by the Weisfeiler-Lehman test. The meaning of "different" here is two graphs are different in their topologies or their properties of interest (i.e., to be predicted)?**
> >
> > **Answer B-2.** Our goal is to distinguish two different graphs that are distinguishable by the WL test, such that two different graphs have **different topologies**or **nonidentical node attributes (labels)**.
> >
> > ---
> >
> > [B] Xu et al. “How Powerful are Graph Neural Networks?.” ICLR 2019.

---

> ### Author Response · Authors · 2020-11-23
> **The end of the discussion phase approaching, and we address all your comments.**
>
> Dear AnonReviewer 3,
>
> We sincerely appreciate your positive comment that we contribute novel ideas that are reasonable.
> During the rebuttal period, we have made every effort to faithfully address all your concerns in the response comment. Based on your comments, we have incorporated more explanations and included more experimental results (Please see all revised captions on Table and Figure, standard deviation on classification (with t-test) and reconstruction tasks in Table 1 and Figure 6, leaderboard results in Table 5). Further, we sincerely appreciate your follow-up question, which we have addressed in the response comment, and we believe our answer clears up your question.
>
> If you have any further questions, please let us know since we can have interactions with you only by this Tuesday (24th). We really appreciate your constructive and helpful comments.
>
> Thanks, Authors

---

> ### Author Response · Authors · 2020-11-24
> **Regarding WL-Test**
>
> We sincerely appreciate your follow-up questions with an insightful discussion. To unknown reason, the response we posted was only visible to us, thus we repost the answer here.
>
> **Question C-1. It is well known that a small structural difference can lead to big differences in properties (sometimes, drug vs poison). Can the W-L test capture this?**
>
> **Answer C-1.** As justified in the WL-test for graph isomorphism test [G], WL test **distinguishes almost all graphs**, specifically called 1-dim WL, for **graph isomorphism test within linear time** (See Theorem 1.1 and Theorem 1.2 of [G]). For this property, WL test is well known to distinguish a broad class of graphs [A, D] for graph isomorphism test, which determines whether two graphs have different connectivities or not. Thus, the WL-test may capture the small structural difference you are referring to, in most cases. However, there are some exceptional corner cases [E] where the WL test generates the same value for two different (non-isomorphic) graphs, and thus it is not perfect. Thus, regarding your question, if the WL-test generates **two different values**for two **given molecular graphs**, then it is guaranteed that they are **different**. However, we cannot guarantee that the two graphs are the same, when the test outputs the same value since there are few corner cases.
>
>
> Yet, existing works on GNNs mostly **theoretically justify their connections to the WL-test**, since there does not exist powerful alternatives for distinguishing graphs in an efficient manner, such as [B] which showed that their **node embedding functions**are as powerful as WL-test. On the other hand, we proposed a **graph-pooling operation**to obtain a global graph-level representation that is **as powerful as the WL-test**such that we can map two distinct graphs into two distinct embeddings. Moreover, we consider the connectivity between nodes, to further incorporate the global structure among the nodes.
>
> Please note that our objective is not to obtain a perfectly accurate graph representation, but is to obtain **more accurate**representations than what **existing pooling methods achieve**. Beyond this theoretical justification, we further show that GMT outperforms **more than 10 existing pooling baselines**, either on graph classification (Table 1 and the table in the common response), reconstruction (Figure 5 and 6), or generation (Figure 8 and Table 3).
>
> ---
>
> [A] Shervashidze et al. “Weisfeiler-Lehman Graph Kernels.” JMLR 2011.
>
> [B] Xu et al. “How Powerful are Graph Neural Networks?.” ICLR 2019.
>
> [C] Weisfeiler & Leman. “THE REDUCTION OF A GRAPH TO CANONICAL FORM AND THE ALGEBRA WHICH APPEARS THEREIN” 1968.
>
> [D] Babai & Kucera. “Canonical labelling of graphs in linear average time.” In Foundations of Computer Science 1979.
>
> [E] Cai et al. “An optimal lower bound on the number of variables for graph identification.” Combinatorica, 1992.
>
> [F] Douglas. “The Weisfeiler-Lehman Method and Graph Isomorphism Testing.” arXiv:1101.5211, 2011.
>
> [G] L. Babai, P. Erdös, S. Selkow. “Random Graph Isomorphism.” SIAM J. Computm., 1980.

---

> ### Author Response · Authors · 2020-11-25
> **The interactive discussion phase will end in less than 7 hours**
>
> Dear Reviewer,
>
> Could you please check our response regarding **WL-Test below**, as well as all other responses to your original comments since we cannot have the interaction with you past the discussion phase deadline that is less than 7 hours away?
> We have done our best to address your comments and clarify factual misunderstandings, and have faithfully reflected your comments in the revision. We thank you so much for your efforts in reviewing our paper as well as responsiveness during the discussion phase.
>
> Best regards,
> Authors

---

### Official Review · AnonReviewer4 · 2020-10-29
**The Review**

**Rating:** 7
**Confidence:** 5

**Review:**

This paper proposes a Transformer-like model: Graph Multiset Transformer to perform the graph pool/aggregation. Overall, the technique part is concrete and clear and the experimental evaluations are comprehensive. The authors also prove the expressive power regarding WL-test. However, several points need to be clarified or addressed.
1.	The idea to utilize the Transformer-like architecture to model the graph neural network (GNN) is not new. Some existing works[1,2] have employed the transformer to enhance the expressive power of GNN. It’s better to add more discussions between GTM and the existing works to highlight its contribution. Meanwhile, several studies [3,4,5,6,7] about graph pooling / self-attention are missing. It’s better to make discussions in the related works. Moreover, if possible, I suggest making the comparison with these methods, especially the recent studies to make the whole experimental results more convincing, e.g. HaarPool, EigenPool, etc.

1.	About the experimental settings. 1) In Section 4.1, the authors state that the 4 molecule datasets are obtained from OGB dataset. However, in OGB dataset, it only contains HIV, while Tox21, Toxcast, BBBP is not included. Maybe there is a mistake. 2) For the molecular dataset, the data splitting is very curial for the final results. Meanwhile, the atom/bond feature extraction process for the molecular datasets is unclear. The authors need to clarify the data splitting (random/scaffold) and feature extraction process to ensure the reproducibility of experiments.

Minor:

In Equation 6, what is the QW_i^Q ?

In Equation 8, why we need MH(H, H, H)? How about directly applying H into SelfAtt block, i.e., Z=H?

In the experiments, this paper only evaluates the memory efficiency of GMT. I would like to see the evaluation about the time efficiency of GMT with other baselines.

Overall, this paper is well written and the experiments look solid. Considering the novelty issue, I think this is a borderline paper. I recommend “Marginally below acceptance threshold ” and would like to see the author's response.

[1] Rong, Yu, et al. "GROVER: Self-supervised Message Passing Transformer on Large-scale Molecular Data." arXiv preprint arXiv:2007.02835 (2020).

[2] Chithrananda, Seyone, Gabe Grand, and Bharath Ramsundar. "ChemBERTa: Large-Scale Self-Supervised Pretraining for Molecular Property Prediction." arXiv preprint arXiv:2010.09885 (2020).

[3] Wang, Yu Guang, et al. "Haar graph pooling." arXiv (2019): arXiv-1909.

[4] Ma, Yao, et al. "Graph convolutional networks with eigenpooling." Proceedings of the 25th ACM SIGKDD International Conference on Knowledge Discovery & Data Mining. 2019.

[5] Bianchi, Filippo Maria, Daniele Grattarola, and Cesare Alippi. "Spectral clustering with graph neural networks for graph pooling." (2020).

[6] Ranjan, Ekagra, Soumya Sanyal, and Partha P. Talukdar. "ASAP: Adaptive Structure Aware Pooling for Learning Hierarchical Graph Representations." AAAI. 2020.

[7] Li, Jia, et al. "Semi-supervised graph classification: A hierarchical graph perspective." The World Wide Web Conference. 2019.

---

> ### Author Response · Authors · 2020-11-14
> **Initial Response to R4 (2/2)**
>
> **Question 3:** In Equation 6, what is the $QW_i^Q$?
>
> **Answer:** $QW_i^Q$ consists of the input query $Q$ and the learnable parameterized weight matrix $W_i^Q$. Therefore, it depends on the input query $Q$, which is $S$ in equation 7 and $H$ in equation 8.
>
> ---
>
> **Question 4:** In Equation 8, why we need MH($H$, $H$, $H$)? How about directly applying $H$ into SelfAtt block, i.e., $Z=H$?
>
> **Answer:** The term MH($H$, $H$, $H$) is not an operation newly introduced with our method, but is the **standard multi-headed attention operation for a self-attention**layer. The conventional attention operation MH($Q$, $K$, $V$) computes the relevance score of $Q$ to $K$, and then performs the weighted sum of $V$ with the calculated relevance scores. For self-attentions, $Q=K=V$, and thus MH($H$, $H$, $H$) first computes the relevance scores of every pair of elements in the input vector $H$, and then performs the weighted sum of elements in $H$ with calculated relevance scores. If we set $Z = H$, then the SelfAtt function becomes a simple linear layer with skip connection, and thus can not consider the inter-node relationships.
>
> ---
>
> **Question 5:** In the experiments, this paper only evaluates the memory efficiency of GMT. I would like to see the evaluation about the time efficiency of GMT with other baselines.
>
> **Answer:** Thanks for your suggestion. We have additionally **provided the results about the time efficiency**of GMT with other baselines in Figure 4 of the revision.
>
> ---
>
> [A] Hu et al. “Open Graph Benchmark: Datasets for Machine Learning on Graphs.” arXiv 2020.
>
> [B] https://ogb.stanford.edu/docs/graphprop/
>
> [C] Niepert et al. “Learning convolutional neural networks for graphs.” ICML 2016.
>
> [D] Zhang et al. “An End-to-End Deep Learning Architecture for Graph Classification.” AAAI 2018.
>
> [E] Xu et al. “How Powerful are Graph Neural Networks?.” ICLR 2019.
>
> [F] Chang and Lin. “Libsvm: A library for support vector machines.” TIST 2011.
>
> [G] Errica et al. “A Fair Comparison of Graph Neural Networks for Graph Classification” ICLR 2020.

---

> > ### Comment · AnonReviewer4 · 2020-11-24
> > **The response**
> >
> > Thank you for your detailed response.  And sorry for the late response.
> > Generally, I'm satisfied with the explanations.
> >
> > For question 1-1, First,
> > > "Conventional self-attention mechanisms can capture a set but not a multiset"
> >
> > This statement is not true.  I think I do not misunderstand your model. Yes, you propose a graph pooling method that considers the nodes as a multiset.  As you mentioned in Sec 3.2, paragraph 2:
> >
> > > "To resolve this issue, we consider attention mechanism on the multiset pooling function to capture structural dependencies
> > among nodes within a graph"
> >
> > Actually, [7] has employed the multi-head self-attention mechanism to model structural dependencies of nodes. From the methodology perspective, you do employ a Transformer-like model to model the multiset pooling. It's better to add more discussions here.
> >
> > For question 3, I understand that. It's better to add this explanation to the revision to make it more clear.
> >
> > For questing 4, yes, in this paper, the main contribution is to employ this multi-head self-attention to model the graph pooling. To confirm the effectiveness of multi-head self-attention, it's better to add the ablation study with this simple linear layer with skip connection.
> >
> > Overall, the work looks solid.  I will increase my score.

---

> > > ### Author Response · Authors · 2020-11-24
> > > **Re: The response.**
> > >
> > > Thank you for your response, we appreciate your helpful comments, and we have further revised our paper, as suggested.
> > >
> > > **Question A-1. The statement, conventional self-attention mechanisms can capture a set but not a multiset, is not true.**
> > >
> > > **Answer A-1.**
> > >
> > > * To be more precise, what you said is true since it is not the fundamental limitation of the self-attention mechanism. However, our intention is that conventional self-attention mechanisms [A, B] did not consider multisets, which have elements with identical feature vectors.
> > >
> > > * We would further like to emphasize that we do not leverage the self-attention layer in the reconstruction task (Figure 5, Figure 6 and Figure 7), although we use it for classification. Thus, what we consider as the essential component for the proposed method is **graph multiset pooling**that condenses the n nodes into the k nodes by considering graph structure into account.
> > >
> > > ---
> > >
> > > **Question A-2. Actually, [7] has employed the self-attention mechanism to model structural dependencies of nodes. It's better to add more discussions here.**
> > >
> > > **Answer A-2.** Thanks for suggesting a relevant work. We want to point out that [7] does not consider the inter-node relationships, as done with our GMT. As shown in the equation 6 of the [7], they use the attention mechanism only for **scoring the importance**of nodes **independently from the other nodes**. Thus there is no **inter-node relationships**captured by the model. We have discussed this point in the related work section of the revision.
> > >
> > > ---
> > >
> > > **Question A-3. I understand Question 3. It's better to add this explanation to the revision to make it more clear.**
> > >
> > > **Answer A-3.** Thank you for the helpful suggestion. We have included this explanation in the graph multi-head attention paragraph of the revision as suggested.
> > >
> > > ---
> > >
> > > **Question A-4. The main contribution is to employ this multi-head self-attention to model the graph pooling. To confirm the effectiveness of multi-head self-attention, it's better to add the ablation study with this simple linear layer with skip connection.**
> > >
> > > **Answer A-4.**
> > > * We want to clarify again that the main contribution is in **compressing the n-nodes over the multiset (Figure 1, right) into the k-nodes**with the **graph multiset pooling function**(GMPool in Figure 2), not the use of self-attention mechanism (SelfAtt in Figure 2). As we mentioned in the previous response, we do not use the self-attention layer for the **graph reconstruction task**.
> > >
> > > * Table 2 of the original paper and the revision contains the suggested ablation study, where we report the performance of the simple linear layer w/o self-attention (w/o the skip-connections). The results show that using self-attention does help with the classification performance. However, in reconstruction tasks (Figure 5, 6, 7), we outperform baselines even without self-attention. We believe this can better show the effectiveness of our method since faithful reconstruction of the original graph is more necessary for reconstruction, than classification which only requires to capture discriminative information.
> > >
> > > ---
> > >
> > > [7] Li, Jia, et al. "Semi-supervised graph classification: A hierarchical graph perspective." The World Wide Web Conference. 2019.
> > >
> > > [A] Vaswani et al. “Attention Is All You Need.” NIPS 2017.
> > >
> > > [B] Lee et al. "Set Transformer: A Framework for Attention-based Permutation-Invariant Neural Networks." ICML 2019.

---

> > > > ### Comment · AnonReviewer4 · 2020-11-24
> > > > **More question about the difference between self-attention and Graph Multiset Pooling**
> > > >
> > > > Thank you for your response.
> > > >
> > > > > We want to clarify again that the main contribution is in compressing the n-nodes over the multiset (Figure 1, right) into the k-nodes with the graph multiset pooling function(GMPool in Figure 2), not the use of self-attention mechanism (SelfAtt in Figure 2).
> > > >
> > > > As you mentioned before, if I do not misunderstand, GMPool tasks the n-nodes as input and produces k-nodes representation. However, the self-attention in [7] can be re-written like:
> > > > $S = \text{sofmax}(W_{s2}tanh(W_{s1} GNN(H, A)))$
> > > >
> > > > Here, $S\in \mathbb{R}^{k \times n}$ which can be viewed as a k-node representation in the latent space. Therefore, I think the claim
> > > >
> > > > > scoring the importance of nodes independently from the other nodes
> > > >
> > > > is not true.
> > > >
> > > > Furthermore, could you give more elaborations about the "inter-node relationships" that you mentioned above?
> > > > Firstly, I'm confused about which component captures inter-node relationships. Are the $GMPool$ or $SelfAtt$?
> > > > Second, In both eq (7) and eq (8), they take $GMH$ and $MH$  as input and consequently apply the LN and rFF on them. Why do you say they can capture inter-node relationships (by LN or rFF ?) while the self-attention can't?

---

> > > > > ### Author Response · Authors · 2020-11-24
> > > > > **Re: More question about the difference between self-attention and Graph Multiset Pooling**
> > > > >
> > > > > We sincerely appreciate your insightful comments. We answer your questions as below:
> > > > >
> > > > > **Question B-1. Scoring the importance of nodes independently from the other nodes in [7] is not true.**
> > > > >
> > > > > **Answer.**
> > > > > - The self-attention in [7] is used to score the n-nodes with k-dimensional perspectives (See equation (6) of [7] with the description on page 4 of section 3.2, which clearly says “k experts to give their opinions about the importance of each node independently”), such that they pool n-nodes into k-nodes with independently calculated k-dimensional importance scores. In other words, equation 6 of [7] $S = \text{softmax}(W_{s2}tanh(W_{s1} H^T))$ does not internally consider the relationships between nodes.
> > > > > - However, as you pointed out, local relationships between **neighboring nodes**in A can be captured from the previously obtained node embedding from the $GNN(H, A)$ term, although it does not utilize the conventional **self-attention**[A] to capture the global relationships between node embeddings in a given graph as a whole. Therefore, we have replaced the term “independently” with “locally” in the related work section of the revision. Thank you for pointing it out.
> > > > >
> > > > > ---
> > > > >
> > > > > **Question B-2-1. Furthermore, could you give more elaborations about the inter-node relationships that you mentioned above? Firstly, I'm confused about which component captures inter-node relationships.**
> > > > >
> > > > > **Answer.** Inter-node relationships are captured by our $SelfAtt$ function in equation 8, as the paragraph name “Self-Attention for Inter-node Relationship” indicates, not by the $GMPool$ in equation 7 of our paper and equation 6 of [7].
> > > > >
> > > > > ---
> > > > >
> > > > > **Question B-2-2. Second, In both eq (7) and eq (8), they take $GMH$ and $MH$ as input and consequently apply the LN and rFF on them. Why do you say they can capture inter-node relationships while the self-attention in [7] can't?.**
> > > > >
> > > > > **Answer.**
> > > > >
> > > > > - Inter-node relationships are captured by the same query, key, and value matrices (i.e. $MH(H, H, H)$ with $Q=K=V=H$ as denoted in equation 8), not by the $GMH$ or $MH$ itself. Specifically, in our $SelfAtt$, we first compute the all pairwise n to n mapping scores with the dot product of the query $H$ with all keys $H$, and then compute the weighted sum of all elements $H$ by calculating the n to n mapping scores (Please see $Att(Q, K, V)$ function in the Graph Multi-head Attention paragraph with the special case $Q=K=V$). Thus, **we can calculate the inter-node scores with the same query and key matrices**, and further consider the inter-node relationships on the value matrices with calculated inter-node scores.
> > > > > - Thus, the proposed $SelfAtt$ can capture the inter-node relationships with a $Q=K=V=H$ condition, and the self-attention in [7] can not consider the inter-node relationships, as mentioned in the answer of Question B-1.
> > > > >
> > > > > ---
> > > > >
> > > > > [7] Li, Jia, et al. "Semi-supervised graph classification: A hierarchical graph perspective." The World Wide Web Conference. 2019.
> > > > >
> > > > > [A] Vaswani et al. “Attention Is All You Need.” NIPS 2017.

---

> > > > > > ### Comment · AnonReviewer4 · 2020-11-25
> > > > > > **Thank you for your explaination.**
> > > > > >
> > > > > > Thank you for your response. I think the thing is more clear now.  You explicitly employ $MH(H, H, H)$ to extract the node-wise interaction.  This is the core difference compared with the previous self-attentive pooling approaches.
> > > > > >
> > > > > > But I still insist that the self-attention in [7] can somehow capture the node-wise interaction in some sense. In [7],  k-dimensional importance scores are not independent since they are projected from a high-dimension latent space.
> > > > > >
> > > > > > I'll consider raising my score during the discussion.

---

> > > > > > > ### Author Response · Authors · 2020-11-25
> > > > > > > **Thank you for your timely response, and we further explain the attention for node-wise interactions.**
> > > > > > >
> > > > > > > We thank you for your timely response with follow-up comments. As you mentioned, we employ $MH(H, H, H)$ to explicitly consider the node-wise interactions.
> > > > > > >
> > > > > > > **Question. But I still insist that the self-attention in [7] can somehow capture the node-wise interaction in some sense. In [7], k-dimensional importance scores are not independent since they are projected from a high-dimension latent space.**
> > > > > > >
> > > > > > > **Answer.**
> > > > > > > We first revisit the self-attention in [7] as follows:
> > > > > > >
> > > > > > > $S = \text{softmax}(W_{s2}tanh(W_{s1} H^T))$.
> > > > > > >
> > > > > > > We agree that what you said is true, to be more precise, since the weight matrix $W_{s1}$ and $W_{s2}$ of self-attention equation in [7] can **implicitly**learn to capture the dependencies by learning how to project the node embeddings from $d$ to $k$ dimension.
> > > > > > >
> > > > > > > However, our intention here is that our self-attention in equation 8 **explicitly**considers the node-wise relationships, even **after compressing the $n$ nodes into the $k$ nodes**with GMPool in Figure 2. For simplicity, we here use the multi-head attention with the single-head ($h=1$) case: $MH(H, H, H) = Att(HW^Q, HW^K, HW^V) W^O$. Then, $Att(HW^Q, HW^K, HW^V) = w(HW^Q (HW^K)^T) HW^V$, where $w$ is an activation function. To consider node-wise interactions,
> > > > > > >
> > > > > > > * we first calculate the $n$-to-$n$ node-wise interactions with the query and key matrices: $w(HW^Q (HW^K)^T)$,
> > > > > > > * and then further compute the weighted sum of their calculated node-wise scores with the value matrix: $w(HW^Q (HW^K)^T) HW^V$,
> > > > > > >
> > > > > > > where the output representation of each node from $Att(HW^Q, HW^K, HW^V)$ can be seen as the convex combination of other nodes. To summarize, the self-attention in [7] can implicitly consider the interactions among nodes with linear projections, while the self-attention used in the GMT can explicitly consider the interactions between nodes with the query, key, and value matrices.
> > > > > > >
> > > > > > > ---
> > > > > > >
> > > > > > > **This is the core difference compared with the previous self-attentive pooling approaches.**
> > > > > > >
> > > > > > > **Contribution.**
> > > > > > > We would like to emphasize that the **self-attention function in equation 8 is merely a single component**of the proposed GMT to consider the relationships between multiset elements. Our main contribution is **not the adaptation of the self-attention schemes**, but the proposal of a graph pooling method that can encode input node embeddings as a multiset, which allows for repeating elements, as well as the consideration of the global connectivities among them.
> > > > > > >
> > > > > > > - 1) We propose a multiset-based pooling method to allow for repeating nodes in a given graph (B. Multiset in Figure 1, right, and Multiset Encoding paragraph in Section 3.2).
> > > > > > >
> > > > > > > - 2) We further define a graph multiset encoding to explicitly consider the graph structure into account (C. Graph Multiset in Figure 1, right, and Graph Multi-head Attention paragraph in Section 3.2).
> > > > > > >
> > > > > > > - 3) We **theoretically justify**that the proposed pooling operation can be as powerful as the WL test, and also can be extended to hierarchical pooling via node clustering.
> > > > > > >
> > > > > > >
> > > > > > > We sincerely appreciate your insightful comments as well as responsiveness during the timely discussion phase. We hope the above responses satisfactorily address your points. Please let us know if there is anything else we should address.
> > > > > > >
> > > > > > > Thanks, Authors
> > > > > > >
> > > > > > > ---
> > > > > > >
> > > > > > > [7] Li, Jia, et al. "Semi-supervised graph classification: A hierarchical graph perspective." The World Wide Web Conference. 2019.

---

> > > > > ### Author Response · Authors · 2020-11-25
> > > > > **The interactive discussion period will end in less than 7 hours.**
> > > > >
> > > > > Dear Reviewer,
> > > > >
> > > > > Could you please check our response below? The interactive discussion phase will end in 7 hours and we will not be able to interact with you past the deadline. We thank you so much for your insightful and constructive comments that made our paper stronger, as well as your responsive feedback during the discussion phase!
> > > > >
> > > > > Best regards,
> > > > > Authors

---

> ### Author Response · Authors · 2020-11-14
> **Initial Response to R4 (1/2)**
>
> We sincerely appreciate your constructive and helpful comments. We initially address all your comments below:
>
> **Question 1-1:** The idea to utilize the Transformer-like architecture to model the graph neural network (GNN) is not new.
>
> **Answer:** This is a critical misunderstanding. We want to clarify that our contribution is not using a Transformer-like architecture for GNN. Conventional self-attention mechanisms can capture a set but not a multiset, nor the global graph structure among the nodes, and such a model is one of our baselines. Our contribution is rather the proposal of a **graph pooling method that considers the nodes as a multiset**such that the pooling becomes as powerful as the WL-test, and further considers the **global graph structure**to better obtain a more accurate graph-level representation. This is clearly described in the abstract and the introduction (Please see Figure 1).
>
> ---
>
> **Question 1-2:** Some existing works [1,2] have employed the transformer in GNNs. It’s better to add more discussions between GMT and the existing works.
>
> **Answer:** Thank you for suggesting related works. However, as mentioned in the previous response, our focus is on encoding the node embeddings as a **multiset**, while also considering the **graph structure**among them, not the use of Transformer in GNNs. Moreover, even our Transformer graph pooling baseline is orthogonal from both [1] and [2], since [1] uses Transformer for node embedding rather than graph pooling, and [2] uses Transformer to encode string representations (i.e. SMILES), not the graphs. We have referenced and discussed this in the related work section of the revised version of the paper.
>
> ---
>
> **Question 1-3:** Several studies [3,4,5,6,7] about graph pooling / self-attention are missing. It’s better to discuss and compare them in the related works.
>
> **Answer:** We believe that we have extensively compared against existing pooling methods, and please note that **MinCutPool [5] you mentioned is already compared**with our method in the experimental section in all experiments (Please see Table 1, 3, and Figure 3, 4, 5, 6, 7, 8). However, we further have discussed and compared more baselines as suggested (See related work section, Table 1, and Figure 3, 4), and provide the initial results that we have obtained thus far in the global response.
>
> ---
>
> **Question 2-1:** OGB dataset only contains HIV, while Tox21, Toxcast, and BBBP are not included.
>
> **Answer:** This is a factual misunderstanding, as **OGB dataset does contain 12 molecule datasets**. Please check **Section 6.1: Molecular graphs in OGB paper [A]**, and their public **Graph Property Prediction description page [B]** for more details. Note that the leaderboard only shows the HIV dataset results since it contains a large number of molecule graphs compared to others.
>
> ---
>
> **Question 2-2:** The authors need to clarify the data splitting (random/scaffold) and feature extraction process to ensure the reproducibility of experiments.
>
> **Answer:** The requested feature extraction process is **already described**in the **Implementation Details on Classification Experiments paragraph of Appendix C.2**, and we have further clarified the dataset preprocessing and splitting settings in the revision. Note that we have **strictly followed the conventional settings**for data splitting and feature extraction. In short, for the TU dataset, we follow the data splitting procedure from [C, D, E] for the 10-fold cross-validation with LIBSVM [F], where one of the 10 subsets are used for the hyperparameter search. For the initial input features, we use the one-hot encoding of their atom types as initial node features in the biochemical datasets, and the one-hot encoding of node degrees as initial node features in the social datasets, as suggested by [G] for fair comparisons. For the OGB dataset, we use the training, validation, and test splits provided with the OGB dataset [A]. Also, we follow exactly the same feature extraction processes for atom and bond types in the OGB dataset [A].

---

> ### Author Response · Authors · 2020-11-23
> **The end of the discussion phase approaching, and we address all your comments.**
>
> Dear AnonReviewer 4,
>
> We sincerely appreciate your positive comments that the technical part is concrete and clear with theoretical justification regarding the WL-test, and the experimental evaluations are comprehensive and solid. During the rebuttal period, we have made every effort to faithfully address all your comments. Here, we briefly summarize the major updates as follows:
>
> * Regarding the novelty issue with more baselines (Question 1), we have clarified that our contribution is the proposal of a graph pooling method that considers the nodes as a multiset, and further discussed and compared more related works (Please see related work section, Table 1, Figure 3, Figure 4, and Figure 6 of the revision).
> * Regarding the experimental setting (Question 2), we have further clarified the dataset preprocessing and splits in the Appendix C.2 of the revision.
> * Regarding the time efficiency experiment, we have additionally included the time comparison results in Figure 4 of the revision.
>
> Could you please go over our full responses and the revision since we can have interactions with you only by this Tuesday (24th)? We really appreciate your insightful and constructive comments.
>
> Thanks, Authors

---

### Author Response · Authors · 2020-11-18
**Summary of updates in the first revision**

We thank all reviewers for their valuable suggestions with effort and time. Here we briefly summarize the major update in the revision, and please refer to the responses of each reviewer for more detailed explanations.

**1. Compare and Discuss more baselines.**

* We have extensively compared against the suggested baselines, and will include the full results in the final revision. Note that we are comparing against a state-of-the-art pooling method (MinCutPool, ICML 2020), which outperforms the suggested baselines. The table below shows the intermediate results for the recently proposed baseline models we have obtained thus far (please note that some baselines are **extremely slow**). We have also discussed them in the **related work**section of the revision, and included their results on graph classification tasks with time and memory efficiency in the revision (Please see related work, Table 1, Figure 3, and Figure 4 of the revision).

| | D&D | PROTEINS | MUTAG | HIV | Tox21 | ToxCast | BBBP | IMDB-B | IMDB-M | COLLAB |
| --- | --- | --- | --- | --- | --- | --- | --- | --- | --- | --- |
| StructPool [1] | 78.45 | **75.16** | 79.5 | 75.85 | 75.43 | 62.17 | 67.01 | 72.06 | 50.23 | 77.27 |
| ASAP [2] | 76.58 | 73.92 | 77.83 | 72.86 | 72.24 | 58.09 | 63.50 | 72.81 | 50.78 | 78.64 |
| EdgePool [3] * | 75.85 | 75.12 | 74.17 | 72.66 | 73.77 | 60.70 | 67.18 | 72.46 | **50.79** | - |
| HaarPool [4] * | - | - | 66.11 | - | - | - | 66.11 | 73.29 | 49.98 | - |
| GMT (Ours) | **78.72** | 75.09 | **83.44** | **77.56** | **77.30** | **65.44** | **68.31** | **73.48** | 50.66 | **80.74** |

* (*) Since EdgePool [3] and HaarPool [4] are extremely slow to train on some datasets that contain a relatively large number of nodes and edges such as D&D and COLLAB (the avg number of nodes for D&D/COLLAB is 284/74 and the avg number of edges is 715/2457), we will report the results every time the experiment is done.

---

**2. Report the standard deviation with the t-test for significance.**

* We have **reported the standard deviations as well as the t-test results in Table 1**of the revision. Regarding the t-test, we set the p-value as 0.05, and highlight the best performance and the comparable performance (p > 0.05) of the best as bold.
* As shown in Table 1, the significance of the proposed Graph Multiset Transformer (GMT) becomes more clear with t-test results, since it **achieves the best or the comparable performance in all classification datasets**. Specifically, the proposed GMT achieves the best performances on 7 datasets among 10 classification datasets, and the comparable results on the remaining 3 datasets, whereas the second best baseline (MinCutPool, ICML 2020) only achieves 2 best performances, and 2 comparable performances among 10 datasets.
* We further have **provided the standard deviation on the graph reconstruction task in Figure 6**of the revision, on which we can directly measure the expressiveness of pooling methods, and we obtain even larger performance gain.

---

**3. Include Time and Memory Efficiency.**

* We have compared more suggested baselines, which obtain decent performances, in the memory efficiency (See Figure 3).
* We have **included the time efficiency results in Figure 4**of the revision. As shown in Figure 4, proposed GMT takes less than (or nearly about) a second even for large graphs, while some baselines, such as HaarPool and EdgePool, take more than a minute.

---

[1] Yuan & Ji. "StructPool: Structured Graph Pooling via Conditional Random Fields." ICLR 2020.

[2] Ranjan et al. "ASAP: Adaptive Structure Aware Pooling for Learning Hierarchical Graph Representations." AAAI 2020.

[3] Diehl, Frederik. "Edge Contraction Pooling for Graph Neural Networks." arXiv:1905.10990.

[4] Wang et al. "Haar Graph Pooling." ICML 2020.

---

### Author Response · Authors · 2020-11-23
**The end of the discussion phase approaching**

Dear Reviewers,

Could you please go over our responses and the revision since we can have interactions with you only by this Tuesday (24th)? We have responded to your comments and faithfully reflected them in the revision, and provided additional experimental results that you have requested. We sincerely thank you for your time and efforts in reviewing our paper, and your insightful and constructive comments.

Thanks, Authors

---

### Decision · Program_Chairs · 2021-01-07
**Final Decision**

**Decision:**

Accept (Poster)

**Comment:**

The paper addresses a very important issue in GNN, the definition of a well-defined pooling function for node aggregation. The proposed Graph Multiset Transformer, although not entirely new, seems to be useful in practice. Issues related to experimental results, as well as problems with presentation, have been solved by the authors's rebuttal, that presented solid experimental results and analysis.
Concerns about the real expressivity of the proposed approach when compared to Weisfeiler-Lehman graph isomorphism test do not affect the contribution delivered by the paper, that seems, at this point, significant.